# The Curse of Unrolling:
# Rate of Differentiating Through Optimization

**Damien Scieur**
Samsung SAIL Montreal
damien.scieur@gmail.com

**Quentin Bertrand**
Mila & Universtié de Montréal
quentin.bertrand@mila.quebec

**Gauthier Gidel**
Mila & Université de Montréal
Canada CIFAR AI Chair
gidelgau@mila.quebec

**Fabian Pedregosa**
Google Research
pedregosa@google.com

## Abstract

Computing the Jacobian of the solution of an optimization problem is a central problem in machine learning, with applications in hyperparameter optimization, meta-learning, optimization as a layer, and dataset distillation, to name a few. Unrolled differentiation is a popular heuristic that approximates the solution using an iterative solver and differentiates it through the computational path. This work provides a non-asymptotic convergence-rate analysis of this approach on quadratic objectives for gradient descent and the Chebyshev method. We show that to ensure convergence of the Jacobian, we can either 1) choose a large learning rate leading to a fast asymptotic convergence but accept that the algorithm may have an arbitrarily long burn-in phase or 2) choose a smaller learning rate leading to an immediate but slower convergence. We refer to this phenomenon as the *curse of unrolling*. Finally, we discuss open problems relative to this approach, such as deriving a practical update rule for the optimal unrolling strategy and making novel connections with the field of Sobolev orthogonal polynomials.

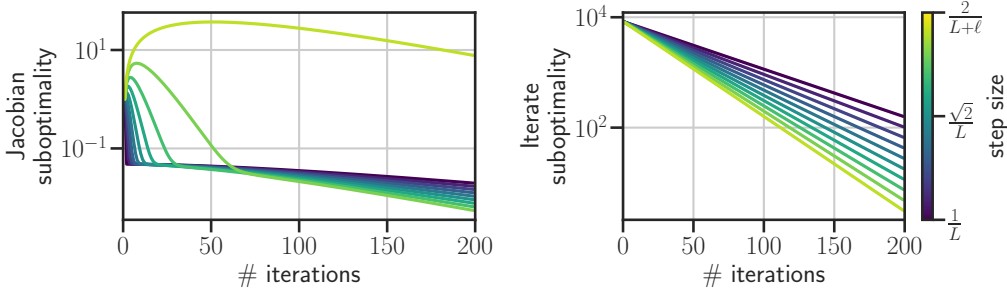

Figure 1: **The Curse of Unrolling:** Better function suboptimality does not imply better Jacobian suboptimality. The speed of the function suboptimality (right) $f(\boldsymbol{x}_t(\boldsymbol{\theta}), \boldsymbol{\theta}) - f(\boldsymbol{x}_\star(\boldsymbol{\theta}), \boldsymbol{\theta})$ of gradient descent is not representative of the speed of convergence of the Jacobian $\|\partial_{\boldsymbol{\theta}} \boldsymbol{x}_t(\boldsymbol{\theta}) - \partial_{\boldsymbol{\theta}} \boldsymbol{x}_\star(\boldsymbol{\theta})\|_F$ (left) at early iterations. The Jacobian suboptimality exhibits a *burn-in* phase where the suboptimality initially increases. The length of this burn-in phase depends on the step-size, with larger step-sizes having smaller function suboptimality but larger burn-in phase.

36th Conference on Neural Information Processing Systems (NeurIPS 2022).

# 1 Introduction

Let $\boldsymbol{x}_\star(\boldsymbol{\theta})$ be a function defined implicitly as the solution to an optimization problem,

$$\boldsymbol{x}_\star(\boldsymbol{\theta}) = \arg\min_{\boldsymbol{x}\in\mathbb{R}^d} f(\boldsymbol{x}, \boldsymbol{\theta}).$$

Implicitly defined functions of this form appear in different areas of machine learning, such as reinforcement learning (Pfau et al., 2016; Du et al., 2017), generative adversarial networks (Metz et al., 2016), hyper-parameter optimization (Bengio, 2000; Pedregosa, 2016; Franceschi et al., 2017a; Lorraine et al., 2020; Bertrand et al., 2020), meta-learning (Franceschi et al., 2018; Rajeswaran et al., 2019), deep equilibrium models, (Bai et al., 2019) or optimization as a layer (Kim et al., 2017; Amos et al., 2017; Wang et al., 2019), to name a few. The main computational burden of using implicit functions in a machine learning pipeline is that the Jacobian computation $\partial_{\boldsymbol{\theta}} \boldsymbol{x}_\star(\boldsymbol{\theta})$ is challenging: since the implicit function $\boldsymbol{x}_\star(\boldsymbol{\theta})$ does not usually admit an explicit formula, classical automatic differentiation techniques cannot be applied directly.

Two main approaches have emerged to compute the Jacobian of implicit functions: *implicit* differentiation and *unrolled* differentiation. This paper focuses on unrolled differentiation while recent surveys on implicit differentiation are (Duvenaud et al., 2020; Blondel et al., 2021).

Unrolled differentiation, also known as iterative differentiation, starts by approximating the implicit function $\boldsymbol{x}_\star(\cdot)$ by the output of an iterative algorithm, which we denote $\boldsymbol{x}_t(\cdot)$, and then differentiates through the algorithm's computational path (Wengert, 1964; Domke, 2012; Deledalle et al., 2014; Franceschi et al., 2017b; Shaban et al., 2019).

**Contributions.** We analyze the convergence of the unrolled Jacobian by establishing worst-case bounds on the Jacobian suboptimality $\|\partial \boldsymbol{x}_t(\boldsymbol{\theta}) - \partial \boldsymbol{x}_\star(\boldsymbol{\theta})\|$ for different methods, more precisely:

1. We provide a general framework for analyzing unrolled differentiation on quadratic objectives for any gradient-based method (Theorem 1). For gradient descent and the Chebyshev iterative method, we derive closed-form worst-case convergence rates in Corollary 1 and Theorem 3.

2. We identify the "curse of unrolling" as a consequence of this analysis: A fast asymptotic rate inevitably leads to a condition number-long burn-in phase where the Jacobian suboptimality increases. While it is possible to reduce the length, or the peak, of this burn-in phase, this comes at the cost of a slower asymptotic rate (Figure 4).

3. Finally, we describe a novel approach to mitigate the curse of unrolling, motivated by the theory of Sobolev orthogonal polynomials (Theorem 4).

**Related work** The analysis of unrolling was pioneered in the work of Gilbert (1992), who showed the asymptotic convergence of this procedure for a class of optimization methods that includes gradient descent and Newton's method. These results have been recently extended by Ablin et al. (2020) and Grazzi et al. (2020), where they develop a complexity analysis for non-quadratic functions. The rate they obtain are valid only for monotone optimization algorithms, such as gradient descent with small step size. We note that Grazzi et al. (2020) developed a non-asymptotic rate for gradient descent that matches our Theorem 2, and provided plots where one can appreciate the burn-in phase, although they did not discuss this behaviour nor the trade-off between the length of this phase and the step-size. Compared to these two papers, we instead focus on the more restrictive quadratic optimization setting. Thanks to this, we obtain tight rates for a larger class of functions, including non-monotone algorithms such as gradient descent with a large learning rate and the Chebyshev method. Furthermore, we also derive novel *accelerated* variants for unrolling (§4.3).

# 2 Preliminaries and Notations

In this paper, we consider an objective function $f$ parametrized by two variables $\boldsymbol{x} \in \mathbb{R}^d$ and $\boldsymbol{\theta} \in \mathbb{R}^k$. We are interested in the derivative of optimization problem solutions:

> **Goal**: approximate Jacobian $\partial \boldsymbol{x}_\star(\boldsymbol{\theta})$, where $\boldsymbol{x}_\star(\boldsymbol{\theta}) = \arg\min_{\boldsymbol{x}\in\mathbb{R}^d} f(\boldsymbol{x}, \boldsymbol{\theta})$. (OPT)

We also assume $\boldsymbol{x}_\star(\boldsymbol{\theta}) \in \mathbb{R}^d$ is the unique minimizer of $f(\boldsymbol{x}, \boldsymbol{\theta})$, for some fixed value of $\boldsymbol{\theta}$. In particular, we will describe the rate of convergence of $\|\partial \boldsymbol{x}_t(\boldsymbol{\theta}) - \partial \boldsymbol{x}_\star(\boldsymbol{\theta})\|$ in the specific case where $f$ is a quadratic function in its first argument, and $\boldsymbol{x}_t$ is generated by a first-order method.

**Notations.** In this paper, we use upper-case letter for polynomials ($P$, $Q$), bold lower-case for vectors ($\boldsymbol{x}$, $\boldsymbol{b}$), and bold upper-case for matrices ($\boldsymbol{H}$). We write $\mathcal{P}_t$ the set of polynomials of degree at least $t$. We distinguish $\nabla f(\boldsymbol{x}, \boldsymbol{\theta})$, that refers to the gradient of the function $f$ in its first argument, and $\partial_{\boldsymbol{\theta}} f(\boldsymbol{x}, \boldsymbol{\theta})$ is the partial derivative of $f$ w.r.t. its second argument, evaluated at $(\boldsymbol{x}, \boldsymbol{\theta})$ (if there is no ambiguity, we write $\partial$ instead of $\partial_{\boldsymbol{\theta}}$). Similarly, $\partial \boldsymbol{x}(\boldsymbol{\theta})$ is the Jacobian of the vector-valued function $\boldsymbol{x}(\cdot)$ evaluated at $\boldsymbol{\theta}$ and $P'(\cdot)$ the derivative of the polynomial $P$. The Jacobian $\partial \boldsymbol{H}(\boldsymbol{\theta})$ a tensor of size $k \times p \times p$. We'll denote its tensor multiplication by a matrix $\boldsymbol{Q} \in \mathbb{R}^{p \times q}$ by $\partial \boldsymbol{H}(\boldsymbol{\theta})\boldsymbol{Q}$, with the understanding that this denotes the multiplication along the first axis, that is, the resulting tensor is characterized by $[\partial \boldsymbol{H}(\boldsymbol{\theta})\boldsymbol{Q}]_i = \partial \boldsymbol{H}(\boldsymbol{\theta})_i \boldsymbol{Q}$ for $1 \le i \le k$. Finally, we denote by $\ell$ the strong convexity constant of the objective function $f$, by $L$ its smoothness constant, and by $\kappa = \ell/L$ its inverse condition number.

## 2.1 Problem Setting and Main Assumptions

Throughout the paper, we make the following three assumptions. The first one assumes the problem is quadratic. The second one is more technical and assumes the Hessian commutes with its derivative respectively, which simplifies considerably the formulas. As we'll discuss in the Experiments section, we believe that some of these assumptions could potentially be relaxed. The third assumption restricts the class of algorithms to first-order methods.

**Assumption 1** (Quadratic objective). *The function $f$ is a quadratic function in its first argument,*

$$f(\boldsymbol{x}, \boldsymbol{\theta}) \overset{def}{=} \tfrac{1}{2} \boldsymbol{x}^\top \boldsymbol{H}(\boldsymbol{\theta})\, \boldsymbol{x} + \boldsymbol{b}(\boldsymbol{\theta})^\top \boldsymbol{x}\,, \tag{1}$$

*where $\ell \boldsymbol{I} \preceq \boldsymbol{H}(\boldsymbol{\theta}) \preceq L\boldsymbol{I}$ for $0 < \ell < L$. We write $\boldsymbol{x}_\star(\boldsymbol{\theta})$ the minimizer of $f$ w.r.t. the first argument.*

**Assumption 2** (Commutativity of Jacobian). *We assume that $\boldsymbol{H}(\boldsymbol{\theta})$ commutes with its Jacobian, in the sense that*

$$\partial \boldsymbol{H}(\boldsymbol{\theta})_i \boldsymbol{H}(\boldsymbol{\theta}) = \boldsymbol{H}(\boldsymbol{\theta})\partial \boldsymbol{H}(\boldsymbol{\theta})_i \ \text{ for } 1 \le i \le k\,. \tag{2}$$

*In the case in which $\boldsymbol{\theta}$ is a scalar ($k = 1$), this condition amounts to the commutativity between matrices $\partial \boldsymbol{H}(\boldsymbol{\theta})\boldsymbol{H}(\boldsymbol{\theta}) = \boldsymbol{H}(\boldsymbol{\theta})\partial \boldsymbol{H}(\boldsymbol{\theta})$*

**Importance.** The previous assumption allows to have simpler expression for the Jacobian of $\boldsymbol{H}^t$. Notably, with this assumption the Jacobian of $\boldsymbol{H}$ can be expressed as

$$\partial \left[ \boldsymbol{H}(\boldsymbol{\theta})^t \right] = t\partial \boldsymbol{H}(\boldsymbol{\theta})\boldsymbol{H}^{t-1}(\boldsymbol{\theta})\,. \tag{3}$$

This assumption is verified for example for Ridge regression (see below). Although quite restrictive, empirical evidence (see Appendix A) suggest that this assumption could potentially be relaxed or even lifted entirely.

**Example 1** (Ridge regression). *Let us fix $\boldsymbol{A} \in \mathbb{R}^{n \times d}, \bar{\boldsymbol{x}} \in \mathbb{R}^d, \boldsymbol{b} \in \mathbb{R}^n$, and let $\boldsymbol{H}(\theta) = \boldsymbol{A}^\top \boldsymbol{A} + \theta \boldsymbol{I}$, where $\theta$ in this case is a scalar. The ridge regression problem*

$$f(\boldsymbol{x}, \theta) = \tfrac{1}{2} \left( \|\boldsymbol{A}\boldsymbol{x} - \boldsymbol{y}\|_2^2 + \theta \|\boldsymbol{x} - \bar{\boldsymbol{x}}\|_2^2 \right),$$

*satisfies Assumptions 1, 2, as $f(\boldsymbol{x}, \theta)$ is quadratic in $\boldsymbol{x}$, and $\partial \boldsymbol{H}(\theta)\boldsymbol{H}(\theta) = \boldsymbol{H}(\theta)\partial \boldsymbol{H}(\theta) = \boldsymbol{H}(\theta)$.*

In our last assumption we restrict ourselves to *first-order method*, widely used in large-scale optimization. This includes methods like gradient descent or Polyak's heavy-ball.

**Assumption 3** (First-order method). *The iterates $\{\boldsymbol{x}_t\}_{t=0\ldots}$ are generated from a first-order method:*

$$\boldsymbol{x}_t(\boldsymbol{\theta}) \in \boldsymbol{x}_0(\boldsymbol{\theta}) + \mathbf{span}\{\nabla f(\boldsymbol{x}_0(\boldsymbol{\theta}), \boldsymbol{\theta}), \ \ldots, \ \nabla f(\boldsymbol{x}_{t-1}(\boldsymbol{\theta}), \boldsymbol{\theta})\}\,.$$

## 2.2 Polynomials and First-Order Methods on Quadratics

The polynomial formalism has seen a revival in recent years, thanks to its simple and constructive analysis (Scieur et al., 2020a; Pedregosa et al., 2020; Agarwal et al., 2021). It starts from a connection between optimization methods and polynomials that allows to cast the complexity analysis of optimization methods as polynomial bounding problem.

### 2.2.1 Connection with Residual Polynomials

When minimizing quadratics, after $t$ iterations, one can associate to any optimization method polynomial $P_t$ of degree at most $t$ such that $P_t(0) = 1$, i.e.,

$$P_t(\lambda) = a_t \lambda^t + \cdots + 1.$$

In such a case, the error at iteration $t$ then can be expressed as

$$\boldsymbol{x}_t(\boldsymbol{\theta}) - \boldsymbol{x}_\star(\boldsymbol{\theta}) = P_t(\boldsymbol{H}(\boldsymbol{\theta}))(\boldsymbol{x}_0(\boldsymbol{\theta}) - \boldsymbol{x}_\star(\boldsymbol{\theta})). \tag{4}$$

This polynomial $P_t(\boldsymbol{H}(\boldsymbol{\theta}))$ is called the *residual polynomial* and represents the output of evaluating the originally real-valued polynomial $P_t(\cdot)$ at the matrix $\boldsymbol{H}$.

**Example 2.** *In the case of gradient descent, the update reads* $\boldsymbol{x}_{t+1} - \boldsymbol{x}^\star = (\boldsymbol{I} - \gamma\boldsymbol{H}(\boldsymbol{\theta}))(\boldsymbol{x}_t - \boldsymbol{x}^\star)$, *which yields the residual polynomial* $P_t(\boldsymbol{H}(\boldsymbol{\theta})) = (\boldsymbol{I} - \gamma\boldsymbol{H}(\boldsymbol{\theta}))^t$.

### 2.2.2 Worst-Case Convergence Bound

From the above identity, one can quickly compute a worst-case bound on the associated optimization method. Using the Cauchy-Schwartz inequality on (4), we obtain

$$\|\boldsymbol{x}_t(\boldsymbol{\theta}) - \boldsymbol{x}_\star(\boldsymbol{\theta})\| \leq \|P_t(\boldsymbol{H}(\boldsymbol{\theta}))\| \, \|\boldsymbol{x}_0(\boldsymbol{\theta}) - \boldsymbol{x}_\star(\boldsymbol{\theta})\|.$$

We are interested in the performance of the first-order method on a class of quadratic functions (see Assumption 1), whose Hessian has bounded eigenvalues. Using the fact that the $\ell_2$-norm of a matrix is equal to its largest singular value, the worst-case performance of the algorithm then reads

$$\|\boldsymbol{x}_t(\boldsymbol{\theta}) - \boldsymbol{x}_\star(\boldsymbol{\theta})\| \leq \max_{\lambda \in [\ell, L]} |P_t(\lambda)| \, \|\boldsymbol{x}_0(\boldsymbol{\theta}) - \boldsymbol{x}_\star(\boldsymbol{\theta})\|. \tag{5}$$

Therefore, the worst-case convergence bound is a function of the polynomial associated with the first-order method, and depends on the bound over the eigenvalue of the Hessian ($\lambda \in [\ell, L]$).

### 2.2.3 Expected Spectral Density and Average-Case Complexity

We recall the average-case complexity framework (Pedregosa et al., 2020; Paquette et al., 2022; Cunha et al., 2022), which provides a finer-grained convergence analysis than the worst-case. This framework is crucial in developing an accelerated method for unrolled differentiation (Section 4).

Instead of considering the worst instance from a class of quadratic functions, average-case analysis considers that functions are drawn *at random* from the class. This means that, in Assumption 1, the matrix $\boldsymbol{H}(\boldsymbol{\theta})$, the vector $\boldsymbol{b}(\boldsymbol{\theta})$ and the initialization $\boldsymbol{x}_0(\theta)$ in Assumption 3 are sampled from some (potentially unknown) probability distributions. Surprisingly, we do not require the knowledge of these distributions, instead, the quantity of interest is the *expected spectral density* $\mu(\lambda)$, defined as

$$\mu(\lambda) \overset{\text{def}}{=} \mathbb{E}_{\boldsymbol{H}(\boldsymbol{\theta})}[\mu_{\boldsymbol{H}(\boldsymbol{\theta})}(\lambda)], \qquad \mu_{\boldsymbol{H}(\boldsymbol{\theta})}(\lambda) \overset{\text{def}}{=} \tfrac{1}{d} \sum_{i=1}^d \delta(\lambda - \lambda_i(\boldsymbol{H}(\boldsymbol{\theta}))). \tag{6}$$

In Equation 6, $\lambda_i(\boldsymbol{H}(\theta))$ is the $i$-th eigenvalue of $\boldsymbol{H}(\theta)$, $\delta(\cdot)$ is the Dirac's delta and $\mu_{\boldsymbol{H}(\boldsymbol{\theta})}(\lambda)$ is the *empirical spectral density* (i.e., $\mu_{\boldsymbol{H}(\boldsymbol{\theta})}(\lambda) = 1/d$ if $\lambda$ is an eigenvalue of $\boldsymbol{H}(\theta)$ and 0 otherwise).

Assuming $\boldsymbol{x}_0(\boldsymbol{\theta}) - \boldsymbol{x}_\star(\boldsymbol{\theta})$ is independent of $\boldsymbol{H}(\boldsymbol{\theta})$,[1] the average-case complexity of the first-order method associated to the polynomial $P_t$ as

$$\mathbb{E}[\|\boldsymbol{x}_t(\boldsymbol{\theta}) - \boldsymbol{x}_\star(\boldsymbol{\theta})\|^2] = \mathbb{E}[\|\boldsymbol{x}_0(\boldsymbol{\theta}) - \boldsymbol{x}_\star(\boldsymbol{\theta})\|^2] \int P_t^2(\lambda) \, \mathrm{d}\mu(\lambda). \tag{7}$$

Here, the term in $P_t$ is algorithm-related, while the term in $\mathrm{d}\mu$ is related to the difficulty of the (distribution over the) problem class. As opposed to the worst-case analysis, where only the *worst* value of the polynomial impacts the convergence rate, the average-case rate depends on the expected value of the squared polynomial $P_t$ over the *whole* distribution. Note that in the previous equation, the expectation is taken over problem instances and not over any stochasticity of the algorithm.

---

[1]This assumption can be removed as in (Cunha et al., 2022) at the price of a more complicated expected spectral density.

# 3 The Convergence Rate of differentiating through optimization

We now analyze the rate of convergence of gradient descent and Chebyshev algorithm (optimal on quadratics). We first introduce the *master identity* (Theorem 1), which draws a link between how well a first-order methods estimates $\partial x_\star(\theta)$, its associated residual polynomial $P_t$, and its derivative $P_t'$.

**Theorem 1** (Master identity). *Under Assumptions 1, 2, 3, let $x_t(\theta)$ be the $t^{th}$ iterate of a first-order method associated to the residual polynomial $P_t$. Then the Jabobian error can be written as*

$$\partial x_t(\theta) - \partial x_\star(\theta) = \big(P_t(H(\theta)) - P_t'(H(\theta))H(\theta)\big)(\partial x_0(\theta) - \partial x_\star(\theta)) \\ + P_t'(H(\theta))\partial_\theta \nabla f(x_0(\theta), \theta). \tag{8}$$

The above identity involves the **derivative** of the residual polynomials and not only the **residual** polynomial, as was the case for minimization (4). This difference is crucial and will result in different rates for the Jacobian suboptimality than classical ones for objective or iterate suboptimality.

For conciseness, our bounds make use of the following shorthand notation

$$G \stackrel{\text{def}}{=} \|\partial_\theta \nabla f(x_0(\theta), \theta)\|_F.$$

## 3.1 Worst-case Rates for Gradient Descent

We consider the fixed-step gradient descent algorithm,

$$x_t(\theta) = x_{t-1}(\theta) - \nabla f(x_{t-1}(\theta), \theta). \tag{9}$$

As mentioned in Example 2, the associated polynomial reads $P_t = (1 - h\lambda)^t$. We can deduce its convergence rate after injecting the polynomial in the master identity from Theorem 1.

**Theorem 2** (Jacobian Suboptimality Rate for Gradient Descent). *Under Assumptions 1, 2, let $x_t(\theta)$ be the $t^{th}$ iterate of gradient descent scheme with step size $h > 0$. Then,*

$$\|\partial x_t(\theta) - \partial x_\star(\theta)\|_F \leq \max_{\lambda \in [\ell, L]} \Big| \underbrace{(1 - h\lambda)^{t-1}}_{\text{exponential decrease}} \big\{ \underbrace{(1 + (t-1)h\lambda)}_{\text{linear increase}} \|\partial x_0(\theta) - \partial x_\star(\theta)\|_F + htG \big\} \Big|.$$

**Discussion.** The bound above is a product of two terms: the first term, $(1 - h\lambda)^{t-1}$, is the convergence rate of gradient descent and decreases exponentially for $h \leq \frac{2}{L+\ell}$, while the second term is increasing in $t$. This results in two distinct phases in training: an initial **burn-in** phase, where the second term dominates and the Jacobian suboptimality might be increasing, followed by an **linear convergence** phase where the exponential term dominates. This phenomenon can be seen empirically, see Figure 1. As an illustration, the next corollary exhibits explicit rates for gradient descent in the two special cases where $h = \frac{1}{L}$ (short steps) and $h = \frac{2}{L+\ell}$ (large steps, maximize asymptotic rate).

**Corollary 1.** *For the step size $h = 1/L$, the rate of Theorem 2 reads*

$$\|\partial x_t(\theta) - \partial x_\star(\theta)\|_F \leq \underbrace{(1 - \kappa)^{t-1}}_{\text{exponential decrease}} \left\{ \underbrace{(1 + \kappa(t-1))}_{\text{linear increase}} \|\partial x_0(\theta) - \partial x_\star(\theta)\|_F + \frac{t}{L} G \right\}.$$

*Assuming $G = 0$, the above bound is monotonically decreasing.*

*If instead we take the worst-case optimal (for minimization) step size $h = 2/(L + \ell)$, then we have*

$$\|\partial x_t(\theta) - \partial x_\star(\theta)\|_F \leq \underbrace{\left(\frac{1-\kappa}{1+\kappa}\right)^{t-1}}_{\text{exponential decrease}} \left\{ \underbrace{|2t - 1|}_{\text{linear increase}} \|\partial x_0(\theta) - \partial x_\star(\theta)\|_F + \frac{2t}{L + \ell} G \right\}.$$

*Moreover, assuming $G = 0$, the maximum of the upper bound over $t$ can go up to*

$$\|\partial x_t(\theta) - \partial x_\star(\theta)\|_F \leq O_{\kappa \to 0} \left(\frac{1}{\kappa} \|\partial x_0(\theta) - \partial x_\star(\theta)\|_F\right) \quad at \quad t \approx \frac{1}{\kappa}.$$

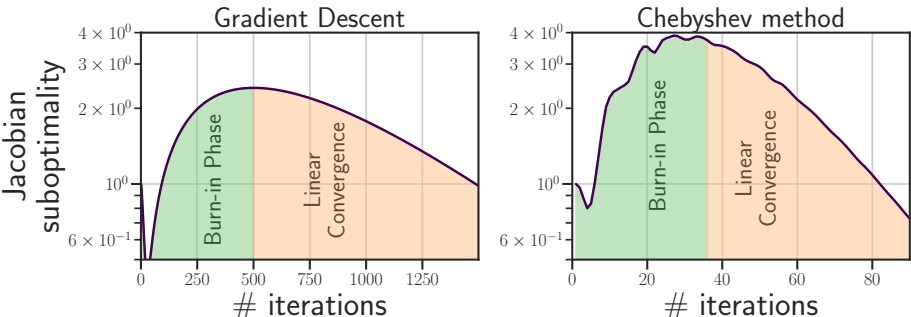

Figure 2: **The Phases of Unrolling**. Gradient descent and the Chebyshev method exhibit two distinct phases during unrolled differentiation: an initial burn-in phase, where the Jacobian suboptimality increases, followed by a convergent phase with an asymptotic linear convergence. The maximum of the suboptimality is similar for both algorithms, but Chebyshev peaks sooner than gradient descent, after a number of iterations equal to (roughly) the square root of the one required by gradient descent. The distinct phases, as well as their relative duration, are predicted by the theoretical worst-case bounds of Corollary 1 and Theorem 3. Both plots are run on the same problem, a ridge regression loss on the `breast-cancer` dataset.

In this Corollary, we see a trade-off between the linear convergence rate (exponential in $t$) and the linear growth in $t$. When the step size is small ($h = 1/L$), the linear rate is slightly slower than the rate of the larger step size ($h = 2/(\ell + L)$). However, the term in $t$ is *way smaller* for the small step size. This makes a big difference in the convergence: for $h = 1/L$, there is *no* local increase, which is not the case for $h = \frac{2}{\ell+L}$. In the next Corollary we provide a bound on the step size $h$ to guarantee a monotone convergence of the Jacobian.

**Corollary 2.** *Assuming $G = 0$, the bound of Theorem 2 is monotonically decreasing for $t \geq 1$ if the step size $h$ from Theorem 2 satisfies $0 < h < \sqrt{2}/L$.*

This bound contrasts with the condition on the step size of gradient descent, which is $h \leq 2/L$, with an optimal value of $h = 2/(\ell + L)$ (Nesterov, 2004). This trade-off between asymptotic rate and length of the burn-in phase leads us to formulate:

---
**The curse of unrolling**

To ensure convergence of the Jacobian with gradient descent, we must either **1)** accept that the algorithm has a burn-in period proportional to the condition number $1/\kappa$, or **2)** choose a small step size that will *slow down* the algorithm's asymptotic convergence.

---

### 3.2   Worst-case Rates for the Chebyshev method

We now derive a convergence-rate analysis for the Chebyshev method, which achieves the best worst-case convergence rate for the minimization of a quadratic function with a bounded spectrum.

**Chebyshev method and Chebyshev polynomials.**   We recall the properties of the Chebyshev method (see e.g. (d'Aspremont et al., 2021, Section 2) for a survey). As mentioned in §2.2.2, the rate of convergence of a first-order method associated with the residual polynomial $P_t$ can be upper bounded by $\max_{\lambda \in [\ell, L]} |P_t(\lambda)|$. Let $\tilde{C}_t$ be the Chebyshev polynomial of the first kind of degree $t$, and define

$$C_t(\lambda) = \frac{\tilde{C}_t(m(\lambda))}{\tilde{C}_t(m(0))}, \qquad m : [\ell, L] \to [0, 1], \quad m(\lambda) = \frac{2\lambda - L - \ell}{L - \ell}. \tag{10}$$

A known property of Chebyshev polynomials is that the *shifted and normalized* Chebyshev polynomial $C_t$ is the residual polynomial with smallest maximum value in the $[\ell, L]$ interval. This implies that the

Chebyshev method, which is the method associated with this polynomial, enjoys the *best* worst-case convergence bound on quadratic functions. Algorithmically speaking, the Chebyshev method reads,

$$\boldsymbol{x}_t(\boldsymbol{\theta}) = \boldsymbol{x}_{t-1}(\boldsymbol{\theta}) - h_t \nabla f(\boldsymbol{x}_{t-1}(\boldsymbol{\theta}), \boldsymbol{\theta}) + m_t(\boldsymbol{x}_{t-1}(\boldsymbol{\theta}) - \boldsymbol{x}_{t-2}(\boldsymbol{\theta})),$$

where $h_t$ is the step size and $m_t$ the momentum. Those parameters are time-varying and depend only on $\ell$ and $L$. The following Proposition shows the rate of convergence of the Chebyshev method.

**Theorem 3** (Jacobian Suboptimality Rate for Chebyshev Method). *Under Assumptions 1,2, let $\xi \overset{def}{=} (1 - \sqrt{\kappa})/(1 + \sqrt{\kappa})$, and $\boldsymbol{x}_t(\boldsymbol{\theta})$ denote the $t^{th}$ iterate of the Chebyshev method. Then, we have the following convergence rate*

$$\|\partial \boldsymbol{x}_t(\boldsymbol{\theta}) - \partial \boldsymbol{x}_\star(\boldsymbol{\theta})\|_F \leq \underbrace{\left( \frac{2}{\xi^t + \xi^{-t}} \right)}_{\text{exponential decrease}} \left\{ \underbrace{\left| \frac{2t^2}{1 - \kappa} - 1 \right|}_{\text{quadratic increase}} \|\partial \boldsymbol{x}_0(\boldsymbol{\theta}) - \partial \boldsymbol{x}_\star(\boldsymbol{\theta})\|_F + \boxed{\frac{2t^2}{L - \ell} G} \right\}.$$

*In short, the rate of the Chebyshev algorithm for unrolling in $O(t^2 \xi^t)$. Moreover, assuming $G = 0$, the maximum of the upper bound over $t$ can go up to*

$$\|\partial \boldsymbol{x}_t(\boldsymbol{\theta}) - \partial \boldsymbol{x}_\star(\boldsymbol{\theta})\|_F \leq O_{\kappa \to 0} \left( \tfrac{2}{\kappa} \|\partial \boldsymbol{x}_0(\boldsymbol{\theta}) - \partial \boldsymbol{x}_\star(\boldsymbol{\theta})\|_F \right) \quad at \quad t \approx 2 \sqrt{\tfrac{1}{\kappa}}.$$

**Discussion.** Despite being optimal for minimization, the rate of the Chebyshev method has an additional $O(t^2)$ factor. Due to this term, the bound diverges at first, similarly to gradient descent with the optimal step size $h = \frac{2}{\ell + L}$, but sooner. This behavior is visible on Figure 2.

# 4 Accelerated Unrolling: How fast can we differentiate through optimization?

We now show to accelerate unrolled differentiation. We first derive a lower bound on the Jacobian suboptimality and then propose a method based on *Sobolev orthogonal polynomials* (Marcellán et al., 2015), which are extremal polynomials for a norm involving both the polynomial and its derivative.

## 4.1 Unrolling is at least as hard as optimization

**Proposition 1.** *Let $\boldsymbol{x}_t$ be the $t$-th iterate of a first-order method. Then, for all iterations $t$ and for all $\boldsymbol{\theta}$, there exists a quadratic function $f$ that verifies Assumption 1 such that $G = 0$, and*

$$\|\partial \boldsymbol{x}_t(\boldsymbol{\theta}) - \partial \boldsymbol{x}_\star(\boldsymbol{\theta})\|_F \geq \frac{2}{\xi^t + \xi^{-t}} \|\partial \boldsymbol{x}_0(\boldsymbol{\theta}) - \partial \boldsymbol{x}_\star(\boldsymbol{\theta})\|_F, \quad \xi = \frac{1 - \sqrt{\kappa}}{1 + \sqrt{\kappa}}. \tag{11}$$

This result tells us that unrolling is *at least* as difficult as optimization. Indeed, the rate in (11) is known to be the lower bound on the accuracy for minimizing smooth and strongly convex function (Nemirovski, 1995). Moreover, although we are not sure if the lower bound is tight for all $t$, we have that when $t \to \infty$, the rate of Chebyshev method matches the above rate.

## 4.2 Average-Case Accelerated Unrolling with Sobolev Polynomials

We now describe an accelerated method for unrolling based on Sobolev polynomials. We first introduce the definition of the Sobolev scalar product for polynomials.

**Definition 1.** *The Sobolev scalar product (and its norm) for two polynomials $P$, $Q$ and a density function $\mu$ is defined as*

$$\langle P, Q \rangle_\eta \overset{def}{=} \int_{\mathbb{R}} P(\lambda) Q(\lambda) \, d\mu + \eta \int_{\mathbb{R}} P'(\lambda) Q'(\lambda) \, d\mu, \quad \|P\|_\eta^2 \overset{def}{=} \langle P, P \rangle_\eta.$$

In the following we'll assume $\mu$ is the *expected spectral density* associated with the current problem class and discuss in the next section some practical choices. Using this scalar product, we can compute a (loose) upper-bound for $\|\partial \boldsymbol{x}_t(\boldsymbol{\theta}) - \partial \boldsymbol{x}_\star(\boldsymbol{\theta})\|_F$ and in Prop. 3, a polynomial minimizing this bound.

**Proposition 2.** *Assume that $\|\partial \boldsymbol{H}(\boldsymbol{\theta})(\boldsymbol{x}_0(\boldsymbol{\theta}) - \boldsymbol{x}_\star(\boldsymbol{\theta}))\|_F \leq \eta \|\partial \boldsymbol{x}_0(\boldsymbol{\theta}) - \partial \boldsymbol{x}_\star(\boldsymbol{\theta})\|_F$. Then, under Assumption 1, 2 and 3, we have the following bound for the average-case rate*

$$\mathbb{E}_{\boldsymbol{H}(\boldsymbol{\theta})}\|\partial \boldsymbol{x}_t(\boldsymbol{\theta}) - \boldsymbol{x}_\star(\boldsymbol{\theta})\|_F^2 \leq 2\|P_t\|_\eta^2 \, \mathbb{E}_{\boldsymbol{H}(\boldsymbol{\theta})}\|\partial \boldsymbol{x}_0(\boldsymbol{\theta}) - \partial \boldsymbol{x}_\star(\boldsymbol{\theta})\|_F^2.$$

**Proposition 3.** *Let $\{S_t\}$ be a sequence of orthogonal Sobolev polynomials, i.e., $\langle S_i, \ S_j \rangle > 0$ if $i = j$ and $0$ otherwise, normalized such that $S_i(0) = 1$. Then, the residual polynomial that minimizes the Sobolev norm can be constructed as*

$$P_t^\star = \underset{P \in \mathcal{P}_t : P(0) = 1}{\arg\min} \langle P, \ P \rangle_\eta = \frac{1}{A_t} \sum_{i=0}^t a_i S_i, \quad \text{where} \quad a_i = \frac{1}{\|S_t\|_\eta^2} \quad \text{and} \quad A_t = \sum_{i=0}^t a_i.$$

*Moreover, we have that $\|P_t^\star\|_\eta^2 = 1/A_t$.*

**Limited burn-in phase.** Using the algorithm associated with $P^\star$ with parameters $\eta$ and $\mu$, we have

$$\mathbb{E}_{\boldsymbol{H}(\boldsymbol{\theta})}\|\partial \boldsymbol{x}_t(\boldsymbol{\theta}) - \partial \boldsymbol{x}_\star(\boldsymbol{\theta})\|_F^2 \leq 2 \cdot \mathbb{E}_{\boldsymbol{H}(\boldsymbol{\theta})}\|\partial \boldsymbol{x}_0(\boldsymbol{\theta}) - \boldsymbol{x}_\star(\boldsymbol{\theta})\|_F^2.$$

This inequality follows directly from Proposition 2 and the optimality of $P_t$ for $\|\cdot\|_\eta$: we have that $\|P_t\|_\eta \leq \|P_{t-1}\|_\eta$ (because $P_{t-1}$ is a feasible solution for $P_t$) and $\|P_t\|_\eta \leq 1$ (because $P_t = 1$ is a feasible solution for any $t > 1$). That is much better than the maximum bump of $(O(1/\kappa)$ from gradient descend (Theorem 1) and from Chebyshev (Theorem 3).

### 4.3 Gegenbaueur-Sobolev Algorithm

In most practical scenarios one does not have access to the expected spectral density $\mu$. Furthermore, the rates of average-case accelerated algorithms have been shown to be robust with respect to distribution mismatch (Cunha et al., 2022). In these cases, we can approximate the expected spectral density by some distribution that has the same support. A classical choice is the Gegenbauer parametric family indexed by $\alpha \in \mathbb{R}$, which encompasses important distributions such as the density associated with Chebyshev's polynomials or the uniform distribution:

$$\mu(\lambda) = \tilde{\mu}(m(\lambda)), \quad \tilde{\mu}(x) = (1 - x^2)^{\alpha - \frac{1}{2}} \quad \text{and} \quad m : [\ell, L] \to [0, 1], \, m(\lambda) = \frac{2\lambda - L - \ell}{L - \ell}. \quad (12)$$

We'll call the sequence of Sobolev orthogonal polynomials for this distribution *Gegenbaueur-Sobolev* polynomials. Although in general Sobolev orthogonal polynomials don't enjoy a three-term recurrence as classical orthogonal polynomials do, for this class of polynomials it's possible to build a recurrence for $S_t$ involving only $S_{t-2}$, $Q_t$ and $Q_{t-2}$, where $\{Q_t\}$ is sequence of Gegenbaueur polynomials (Marcellán et al., 1994). Unfortunately, existing work on Gegenbaueur and Gegenbaueur-Sobolev polynomials considers the un-shifted distribution $\tilde{\mu}$, but not $\mu$, and also doesn't consider the residual normalization $Q_t(0) = 1$ or $S_t(0) = 1$. After (painful) changes to shift and normalize the polynomials, we obtain a three-stages algorithm, summarized in the next Theorem.

**Theorem 4** (Accelerated Unrolling). *Let $P_t^\star$ be defined in Proposition 3, where the Sobolev product is defined with the density function $\mu$ (12). Then, the optimization algorithm associated to $P_t^\star$ reads*

$$\boldsymbol{y}_t = \boldsymbol{y}_{t-1} - h_t \nabla f(\boldsymbol{y}_{t-1}, \boldsymbol{\theta}) + m_t(\boldsymbol{y}_{t-1} - \boldsymbol{y}_{t-2}) \tag{13}$$

$$\boldsymbol{z}_t = c_t^{(1)} \boldsymbol{z}_{t-2} + c_t^{(2)} \boldsymbol{y}_t - c_t^{(3)} \boldsymbol{y}_{t-2} \tag{14}$$

$$\boldsymbol{x}_t = \frac{A_{t-1}}{A_t} \boldsymbol{x}_{t-1} + \frac{a_t}{A_t} \boldsymbol{z}_t, \tag{15}$$

*for some step size $h_t$, momentum $m_t$, parameters $c_t^{(1)}$, $c_t^{(2)}$, $c_t^{(3)}$ that depend on $\alpha$, $\ell$, $L$ and $\eta$, and $A_t$, $a_t$ are defined in Proposition 3, whose recurrence are detailed in Appendix D. Moreover, when $t \to \infty$, the recurrence simplifies into*

$$\boldsymbol{y}_t = \boldsymbol{y}_{t-1} + h \nabla f(\boldsymbol{y}_{t-1}, \boldsymbol{\theta}) + m(\boldsymbol{y}_{t-1} - \boldsymbol{y}_{t-2}) \tag{16}$$

$$\boldsymbol{x}_t = \boldsymbol{y}_t + m(\boldsymbol{x}_{t-1} - \boldsymbol{y}_{t-2}), \tag{17}$$

*where $m = \left(\frac{1 - \sqrt{\kappa}}{1 + \sqrt{\kappa}}\right)^2$ and $h = \left(\frac{2}{\sqrt{\ell} + \sqrt{L}}\right)^2$ are the momentum and step size of Polyak's Heavy Ball. Moreover, as $t \to \infty$, we have the same asymptotic linear convergence as the Chebyshev method,* $\lim \sqrt[t]{\frac{\|\partial \boldsymbol{x}_t(\boldsymbol{\theta}) - \partial \boldsymbol{x}_\star(\boldsymbol{\theta})\|_F}{\|\partial \boldsymbol{x}_0(\boldsymbol{\theta}) - \partial \boldsymbol{x}_\star(\boldsymbol{\theta})\|_F}} \leq \sqrt{m}$.

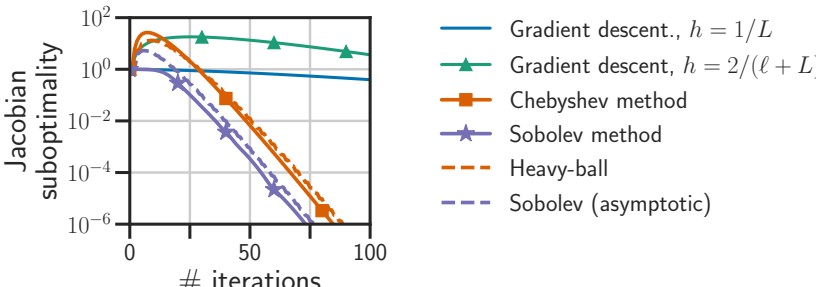

Figure 3: (Theoretical) **Worst-case convergence rates** for different algorithms, with $\ell = 0.5$, $L = 10$, $\alpha = 1$ and $G = 0$. The upper bound of a method associated with the polynomials $\{P_t\}$ is defined as $\max_{\lambda \in [\ell, L]} |P_t(\lambda)|$, where $t$ is the iteration counter. The plot compares Gradient descent with large $(\frac{2}{L+\mu})$ and small $(\frac{1}{L})$ step size, Chebyshev, Sobolev (with $\eta = L/\ell$), and their asymptotic variants. We recognize in those curves the peaks of gradient descent and Chebyshev from Figure 2.

The accelerated algorithm for unrolling is divided into three parts. First, (13) corresponds to an algorithm whose associated polynomials are Gegenbaueur polynomials. This is expected, as Pedregosa et al. (2020) identified that all average-case optimal methods take the form of gradient descent with momentum. Second, (14) builds the Sobolev polynomial that corresponds to a weighted average of $\boldsymbol{y}_t$. Finally, (15) is the weighted average of Sobolev polynomials that builds $P^\star$ in Proposition 2.

The non-asymptotic algorithm is rather complicated to implement; see Appendix D. Moreover, it requires a bound on the spectrum of $\boldsymbol{H}(\boldsymbol{\theta})$, namely $[\ell, L]$, and one also has to choose an associated expected spectral density $\mu(\lambda)$ (parametrized by $\alpha$) and the parameter $\eta$. Nevertheless, this is the method that achieves the best performance for problems that satisfy our assumptions.

Surprisingly, the *asymptotic* version is extremely simple, as it corresponds to *a weighed average of Heavy-Ball iterates*: the only required parameters are $\ell$ and $L$. It means that asymptotically, the algorithm is *universally* optimal, i.e., it achieves the best performing rate as long as we can identify the bounds on the spectrum of $\boldsymbol{H}(\boldsymbol{\theta})$. Such universal properties have been identified previously in (Scieur et al., 2020b), who showed that all average-case optimal algorithms converge to Polyak's momentum independently of the expected spectral density $\mu$ (up to mild assumptions). We have the same phenomenon here, but with the additional surprising (and counter-intuitive) feature that the asymptotic algorithm is also *independent of $\eta$*.

## 5 Experiments and Discussion

### 5.1 Experiments on least squares objective

We compare multiple algorithms for estimating the Jacobian (OPT) of the solution of a ridge regression problem (Example (1)) for a fixed value of $\theta = 10^{-3}$. Figure 1 shows the objective and Jacobian suboptimality on a ridge regression problem with the breast-cancer[2] as underlying dataset. Figure 4 shows the Jacobian suboptimality as a function of the number of iterations, on both the breast-cancer and bodyfat[3] dataset, and for a synthetic dataset (where $\boldsymbol{H}(\boldsymbol{\theta})$ is generated as $\boldsymbol{A}^\top \boldsymbol{A}$, where each entry in $\boldsymbol{A}$ is generated from a standard Gaussian distribution). Appendix B contains further details and experiments on a logistic regression objective.

We observe the early suboptimality increase of Gradient descent and Chebyshev algorithm as predicted by Theorem 2 and Theorem 3. Compared to Figure 3, that showed the theoretical rates, we see that there's a remarkable agreement between theory and practice, as both the early increase, the asymptotic rate and the ordering of the methods matches the theoretical prediction. We also see that Sobolev is the best performing algorithm in practice, as it avoids the early increase while matching the accelerated asymptotic rate of Chebyshev.

---

[2] https://archive.ics.uci.edu/ml/datasets/Breast+Cancer+Wisconsin+(Diagnostic)
[3] http://lib.stat.cmu.edu/datasets/

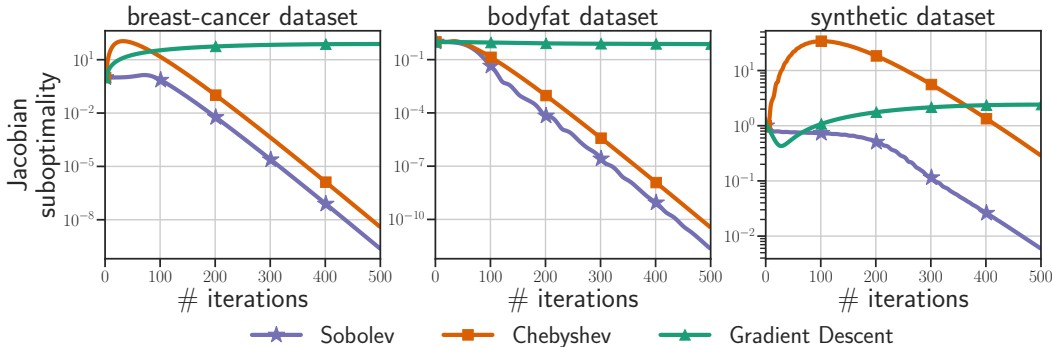

Figure 4: **Empirical comparison** of the Sobolev method introduced in §4.3 (with $\alpha = 1$ and $\eta = 1$), the Chebyshev method and Gradient descent on 3 different datasets. The Sobolev algorithm has the shortest burn-in phase, does not locally diverge and has an accelerated asymptotic rate of convergence.

## 5.2 Experiments on logistic regression objective

In this section we provide some extra experiments on a non-quadratic objective. We choose the following regularized logistic regression objective

$$f(\boldsymbol{x}, \theta) = \sum_{i=1}^{n} \varphi(\boldsymbol{A}_i^{\top} \boldsymbol{x}, \mathrm{sign}(\boldsymbol{b})) \tag{18}$$

where $\varphi$ is the binary logistic loss, $\boldsymbol{A}, \boldsymbol{b}$ is the data, which we generated both from a synthetic dataset and the breast-cancer dataset as described in §5.

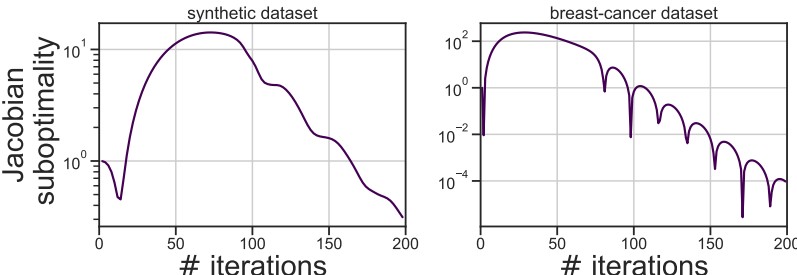

Figure 5: **Two-phase dynamics in logistic regression.** The two-phase dynamics predicted by Corollary 1 and Theorem 3 empirically hold for a logistic regression objective. This objective not covered by our theory since it would violate the quadratic assumption (Assumption 1).

The only significant difference with the least squares loss is the range of step-size values that exhibit the initial burn-in phase. While for the quadratic loss, these are step-sizes close to $2/(L + \mu)$, in the case of logistic regression, L is a crude upper bound and so this step-size is not necessarily the one that achieves the fastest convergence rate. The featured two-phase curve was computed using the step-size with a fastest asymptotic rate, computed through a grid-search on the step-size values.

**Limitations.** Our theoretical results are limited to first-order methods applied to quadratic functions. Many applications use first-order methods, the quadratic Assumption 1, as well as the commutativity Assumption 2, are somewhat restrictive. However, experiments on objectives violating the non-quadratic and non-commutative assumption (Appendix B and A) show that the two-phase dynamics empirically translate to more general objectives. The Sobolev algorithm developed in this paper, however, might not generalize well outside the scope of quadratics. Nevertheless, the development of this accelerated method for unrolling highlights that we can adapt the design of current optimization algorithms so that they might perform better for automatic differentiation.

**Acknowledgements.** The authors would like to thank Pierre Ablin, Riccardo Grazzi, Paul Vicol, Mathieu Blondel and the anonymous reviewers for feedback on this manuscript. QB would like to thank Samsung Electronics Co., Ldt. for funding this research.

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
