# Appendices

## A    On the Commutativity Assumption

We consider the problem

$$f(\boldsymbol{x}, \theta) = \tfrac{1}{2} \left( \|\boldsymbol{A}\boldsymbol{x} - \boldsymbol{y}\|_2^2 + \theta\|\boldsymbol{x} - \bar{\boldsymbol{x}}\|_{\boldsymbol{D}}^2 \right), \text{with } \|\boldsymbol{x}\|_{\boldsymbol{D}}^2 \overset{\text{def}}{=} \boldsymbol{x}^\top \boldsymbol{D}\boldsymbol{d},$$

which is a generalization of Example 1 for the matrix norm $\|\boldsymbol{x}\|_{\boldsymbol{D}}^2$ with a diagonal matrix $\boldsymbol{D}$. Contrary to Example 1, the matrix $\boldsymbol{D}$ is not an identity matrix, but instead a diagonal matrix where the diagonal entries are generated from a Chi-squared distribution. In this case, Assumption 2 is no longer verified.

To investigate whether the two phases dynamics appear also on this class of problems, we repeat the same experiment as in Figure 2 with the above objective. We plot the result here below, confirming the same dynamics of an initial Burn-in-Phase followed by a linear convergence phase observed in the initial experiment.

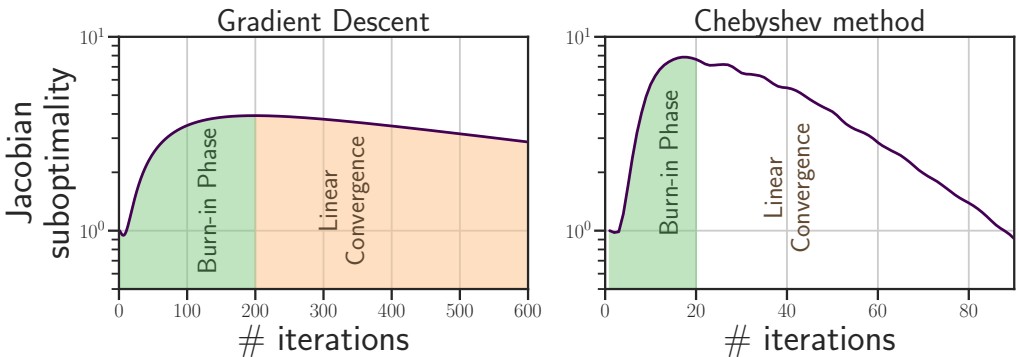

Figure 6: **Two-phase dynamics without the commutativity assumption.** The two-phase dynamics predicted by Corollary 1 and Theorem 3 empirically hold for a problem that does not satisfy the commutativity assumption (Assumption 2).

We also reproduced the same setup as in Figure 4 with this matrix norm, obtaining again comparable results as in the commutative case. This suggest that results regarding the two-phase dynamics could potentially be developed without Assumption 2, as we observe similar results as in Figure 4.

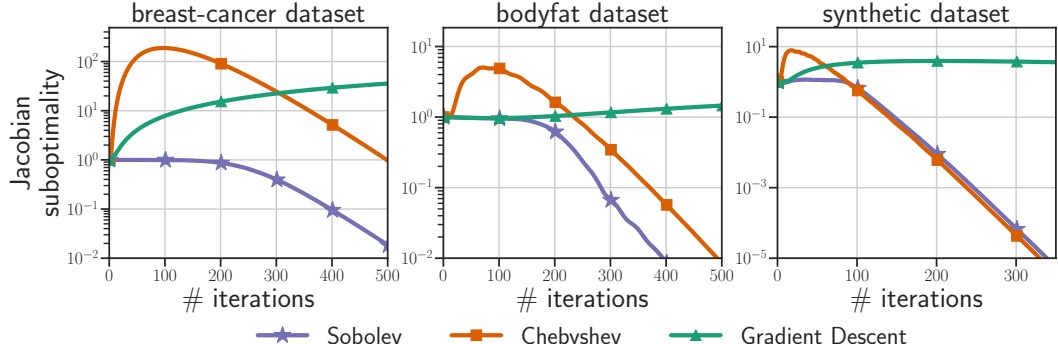

## B    Experiments

### B.1    Further experimental details

**Hyperparameters.**    Initialization is always zero, $\boldsymbol{x}_0 = \boldsymbol{0}$, the regularization parameter $\theta$ in the ridge regression problem is always set to $\lambda = 10^{-3}\|\boldsymbol{A}\|_2$.

| DATASET | $n$ | $d$ | $\kappa$ |
|---|---|---|---|
| BREAST CANCER | 683 | 10 | $7.2 \times 10^7$ |
| BODYFAT | 252 | 14 | 0.021 |
| SYNTHETIC | 200 | 100 | 0.18 |

**Train-test split.**  For every dataset, we only use the train set, where the split is given by the libsvmtools[4] project.

**Run-time.**  Given the reduced size of these datasets, the script to compare all methods, which does a full unrolling for each iteration, runs in under 5 minutes running on CPU.

## C  Proofs

### C.1  Proof of Theorem 1

**Theorem 1** (Master identity). *Under Assumptions 1, 2, 3, let $\boldsymbol{x}_t(\boldsymbol{\theta})$ be the $t^{th}$ iterate of a first-order method associated to the residual polynomial $P_t$. Then the Jabobian error can be written as*

$$\partial \boldsymbol{x}_t(\boldsymbol{\theta}) - \partial \boldsymbol{x}_\star(\boldsymbol{\theta}) = \big(P_t(\boldsymbol{H}(\boldsymbol{\theta})) - P'_t(\boldsymbol{H}(\boldsymbol{\theta}))\boldsymbol{H}(\boldsymbol{\theta})\big)(\partial \boldsymbol{x}_0(\boldsymbol{\theta}) - \partial \boldsymbol{x}_\star(\boldsymbol{\theta}))$$
$$+ P'_t(\boldsymbol{H}(\boldsymbol{\theta}))\partial_{\boldsymbol{\theta}} \nabla f(\boldsymbol{x}_0(\boldsymbol{\theta}), \boldsymbol{\theta}) . \tag{8}$$

*Proof.* We differentiate both sides of (4) and use Assumption 2:

$$\partial \boldsymbol{x}_t(\boldsymbol{\theta}) - \partial \boldsymbol{x}_\star(\boldsymbol{\theta}) = P_t(\boldsymbol{H}(\boldsymbol{\theta}))(\partial \boldsymbol{x}_0(\boldsymbol{\theta}) - \partial \boldsymbol{x}_\star(\boldsymbol{\theta})) + P'(\boldsymbol{H}(\boldsymbol{\theta}))\partial \boldsymbol{H}(\boldsymbol{\theta})(\boldsymbol{x}_0(\boldsymbol{\theta}) - \boldsymbol{x}_\star(\boldsymbol{\theta})) .$$

We now differentiate the equation $\boldsymbol{b}(\boldsymbol{\theta}) = \boldsymbol{H}(\boldsymbol{\theta})\boldsymbol{x}_\star(\boldsymbol{\theta})$ w.r.t. $\boldsymbol{\theta}$,

$$\partial \boldsymbol{b}(\boldsymbol{\theta}) = \partial \boldsymbol{H}(\boldsymbol{\theta})\boldsymbol{x}_\star(\boldsymbol{\theta}) + \boldsymbol{H}(\boldsymbol{\theta})\partial \boldsymbol{x}_\star(\boldsymbol{\theta}).$$

We first substitute $\partial \boldsymbol{H}(\boldsymbol{\theta})\boldsymbol{x}_\star(\boldsymbol{\theta})$ by $\partial \boldsymbol{b}(\boldsymbol{\theta}) - \boldsymbol{H}(\boldsymbol{\theta})\partial \boldsymbol{x}_\star(\boldsymbol{\theta})$. After rearrangement, we finally get

$$\partial \boldsymbol{x}_t(\boldsymbol{\theta}) - \partial \boldsymbol{x}_\star(\boldsymbol{\theta}) = (P_t(\boldsymbol{H}(\boldsymbol{\theta})) - P'(\boldsymbol{H}(\boldsymbol{\theta}))\boldsymbol{H}(\boldsymbol{\theta}))(\partial \boldsymbol{x}_0(\boldsymbol{\theta}) - \partial \boldsymbol{x}_\star(\boldsymbol{\theta}))$$
$$+ P'(\boldsymbol{H}(\boldsymbol{\theta})))[\partial \boldsymbol{H}(\boldsymbol{\theta})\boldsymbol{x}_0(\boldsymbol{\theta}) + \partial \boldsymbol{b}(\boldsymbol{\theta}) + \boldsymbol{H}(\boldsymbol{\theta})\partial \boldsymbol{x}_0(\boldsymbol{\theta})]$$

It suffices to notice that the terms inside the square brackets are the cross-derivative of $f$:

$$\partial_{\boldsymbol{\theta}} \nabla f(\boldsymbol{x}, \boldsymbol{\theta}) = \boldsymbol{H}(\boldsymbol{\theta})\partial \boldsymbol{x}(\boldsymbol{\theta}) + \partial \boldsymbol{H}(\boldsymbol{\theta})\boldsymbol{x}(\boldsymbol{\theta}) + \partial \boldsymbol{b}(\boldsymbol{\theta}).$$

$\square$

### C.2  Proof of Corollary 2

**Corollary 2.** *Assuming $G = 0$, the bound of Theorem 2 is monotonically decreasing for $t \geq 1$ if the step size $h$ from Theorem 2 satisfies $0 < h < \sqrt{2}/L$.*

*Proof.* In this proof, we assume that $t \geq 1$. Indeed, when $t = 0$ and $t = 1$, the worst-case bound do not guarantee any progress over $\|\partial \boldsymbol{x}_1(\boldsymbol{\theta}) - \partial \boldsymbol{x}_\star(\boldsymbol{\theta})\|_F$.

First, we notice that when $h\lambda \leq 1$ (i.e., $h \leq 1/L$), we have that the rate from Theorem 2 is monotonically decreasing. Indeed, the derivative over $t$ gives

$$(1 - h\lambda)^{t-1}((h\lambda(t-1) + 1)\log(1 - h\lambda) + h\lambda).$$

If the following condition is satisfied for all $t \geq 1$, the derivative is negative, and therefore the bound is monotonically decreasing:

$$\log(1 - h\lambda) \leq \frac{h\lambda}{(h\lambda(t-1) + 1)}.$$

---

[4] https://www.csie.ntu.edu.tw/~cjlin/libsvmtools/datasets/

This is always true since the right-hand side is negative, because $h\lambda < 1$, and the left-hand side is always positive since $t \geq 1$.

We now assume that there exist some values of $\lambda$ such that $h\lambda > 1$. For those values of $h\lambda$, the expression in Theorem 2 becomes

$$(h\lambda - 1)^{t-1} \left\{ (1 + (t-1)h\lambda) \|\partial x_0(\theta) - \partial x_\star(\theta)\|_F.$$

We now compute its maximum value. First, we compute its derivative over $t$ and solve $\frac{\mathrm{d}\cdot}{\mathrm{d}t} = 0$. We obtain the unique solution

$$t_\star = 1 - \frac{1}{\log(h\lambda - 1)} - \frac{1}{h\lambda}.$$

This means there is only one maximum in the expression. We now seek a value of $h\lambda$ where the bound decrease monotonically for $t > 1$, i.e.,

$$\|\partial x_1(\theta) - \partial x_\star(\theta)\|_F > \|\partial x_2(\theta) - \partial x_\star(\theta)\|_F > \|\partial x_3(\theta) - \partial x_\star(\theta)\|_F > ...$$

Since we know there is only one maximum, we compute $h\lambda$ such that, in the worst case, $\|\partial x_1(\theta) - \partial x_\star(\theta)\|_F = \|\partial x_2(\theta) - \partial x_\star(\theta)\|_F$. We therefore have to solve

$$(h\lambda - 1)(1 + h\lambda) = 1 \quad \Rightarrow \quad h\lambda = \sqrt{2}.$$

In particular, this means that if $h\lambda < \sqrt{2}$, the bound decreases monotonically for $t = 1, 2, \dots$.

$\square$

## C.3 Proof of Theorem 3

**Theorem 3** (Jacobian Suboptimality Rate for Chebyshev Method). *Under Assumptions 1,2, let $\xi \overset{def}{=} (1 - \sqrt{\kappa})/(1 + \sqrt{\kappa})$, and $x_t(\theta)$ denote the $t^{th}$ iterate of the Chebyshev method. Then, we have the following convergence rate*

$$\|\partial x_t(\theta) - \partial x_\star(\theta)\|_F \leq \underbrace{\left(\frac{2}{\xi^t + \xi^{-t}}\right)}_{\text{exponential decrease}} \left\{ \underbrace{\left|\frac{2t^2}{1-\kappa} - 1\right|}_{\text{quadratic increase}} \|\partial x_0(\theta) - \partial x_\star(\theta)\|_F + \frac{2t^2}{L-\ell} G \right\}.$$

*In short, the rate of the Chebyshev algorithm for unrolling in $O(t^2\xi^t)$. Moreover, assuming $G = 0$, the maximum of the upper bound over $t$ can go up to*

$$\|\partial x_t(\theta) - \partial x_\star(\theta)\|_F \leq O_{\kappa \to 0}\left(\frac{2}{\kappa}\|\partial x_0(\theta) - \partial x_\star(\theta)\|_F\right) \quad at \quad t \approx 2\sqrt{\frac{1}{\kappa}}.$$

*Proof.* First, we recall that the derivative of the Chebyshev polynomial of the first kind can be expressed as a function of the Chebyshev polynomial of the second kind (written $\tilde{U}_t$):

$$\frac{\mathrm{d}\tilde{C}_t(\lambda)}{\mathrm{d}\lambda} = t\tilde{U}_{t-1}(\lambda).$$

Therefore, we replace the polynomial $P$ in Theorem 1 by $C_t$, and evaluate

$$C_t(\lambda) - \lambda\frac{\mathrm{d}C_t(\lambda)}{\mathrm{d}\lambda} = C_t(\lambda) - \lambda\frac{m'(\lambda)\tilde{C}_t'(m(\lambda))}{\tilde{C}_t(m(0))} = C_t(\lambda) - \frac{2\lambda t\tilde{U}_{t-1}(m(\lambda))}{(L-\ell)\tilde{C}_t(m(0))}.$$

This polynomial achieves its maximum in absolute value at the end of the interval $[\ell, L]$. Therefore, after replacement, and using the fact that $m(L) = 1$, $\tilde{C}(1) = 1$, and $\tilde{U}_t(1) = t$, we obtain

$$\left|\left[C_t(\lambda) - \frac{2\lambda t\tilde{U}_{t-1}(m(\lambda))}{(L-\ell)\tilde{C}_t(m(0))}\right]_{\lambda=L}\right| = \frac{1}{|\tilde{C}(m(0))|}\left|\frac{2t^2}{1-\kappa} - 1\right|.$$

Similarly, for the second term, we have

$$\max_{\lambda \in [\ell, L]} \frac{\mathrm{d}C_t(\lambda)}{\mathrm{d}\lambda} = \frac{1}{|\tilde{C}(m(0))|}\frac{2t^2}{1-\kappa}.$$

It suffices now to evaluate $\frac{1}{|\tilde{C}(m(0))|}$. Using (for example) (d'Aspremont et al., 2021, Theorem 2.1), we finally have

$$\frac{1}{|\tilde{C}(m(0))|} = \frac{1}{\xi^t + \xi^{-t}}, \qquad \xi = \frac{1 - \sqrt{\kappa}}{1 + \sqrt{\kappa}}.$$

$\square$

## C.4 Proof of Proposition 1

**Proposition 1.** *Let $\boldsymbol{x}_t$ be the $t$-th iterate of a first-order method. Then, for all iterations $t$ and for all $\boldsymbol{\theta}$, there exists a quadratic function $f$ that verifies Assumption 1 such that $G = 0$, and*

$$\|\partial \boldsymbol{x}_t(\boldsymbol{\theta}) - \partial \boldsymbol{x}_\star(\boldsymbol{\theta})\|_F \geq \frac{2}{\xi^t + \xi^{-t}} \|\partial \boldsymbol{x}_0(\boldsymbol{\theta}) - \partial \boldsymbol{x}_\star(\boldsymbol{\theta})\|_F, \quad \xi = \frac{1 - \sqrt{\kappa}}{1 + \sqrt{\kappa}}. \tag{11}$$

*Proof.* The proof is based on a reduction to the optimization case. Indeed, consider the specific case of ridge regression, with a free scaling parameter $\alpha > 0$,

$$f(\boldsymbol{x}, \boldsymbol{\theta}) = \frac{1}{2} \left( \|\boldsymbol{A}\boldsymbol{x} - \boldsymbol{b}\|^2 + \alpha\boldsymbol{\theta}\|\boldsymbol{x} - \boldsymbol{x}_0\|^2 \right).$$

In such a case, for all $\boldsymbol{x}_0$, we have $\|\partial_{\boldsymbol{\theta}} \nabla f(\boldsymbol{x}_0(\boldsymbol{\theta}), \boldsymbol{\theta})\|_F = 0$. Moreover, this function is $[\sigma^2_{\min}(\boldsymbol{A}) + \alpha\boldsymbol{\theta}]$ strongly convex and $[\sigma^2_{\max}(\boldsymbol{A}) + \alpha\boldsymbol{\theta}]$-smooth, where $\sigma_{\min}$ and $\sigma_{\max}$ are respectively the smallest and largest singular value of a matrix. Let us write $\boldsymbol{H} = \boldsymbol{A}^\top \boldsymbol{A} + \alpha\boldsymbol{\theta}\boldsymbol{I}$ and $\boldsymbol{x}_\star = \boldsymbol{H}^{-1}(\boldsymbol{\theta})\boldsymbol{A}^T b$.

Now, consider any quadratic function $\tilde{f}$ of the form

$$\tilde{f} = \frac{1}{2}(\boldsymbol{x} - \tilde{\boldsymbol{x}}_\star)\tilde{\boldsymbol{H}}(\boldsymbol{x} - \tilde{\boldsymbol{x}}_\star).$$

Using the notation $\bar{\boldsymbol{\theta}}$ to be a *fixed* value of theta $\boldsymbol{\theta}$ (i.e., $\bar{\boldsymbol{\theta}} = \boldsymbol{\theta}$ but $\partial_{\boldsymbol{\theta}} \bar{\boldsymbol{\theta}} = 0$), it is possible to write $f$ such that it matches $\tilde{f}$, by setting

$$\boldsymbol{A} = (\tilde{\boldsymbol{H}} - \alpha\bar{\boldsymbol{\theta}})^{\frac{1}{2}}, \quad b = \boldsymbol{A}(\boldsymbol{A}^\top \boldsymbol{A})^{-1}(\boldsymbol{A}^\top \boldsymbol{A} + \alpha\bar{\boldsymbol{\theta}}\boldsymbol{I})\tilde{\boldsymbol{x}}_\star.$$

This is possible only if $\tilde{\boldsymbol{H}} - \alpha\bar{\boldsymbol{\theta}} \succ \boldsymbol{0}$, or equivalently, if $\ell > \alpha\bar{\boldsymbol{\theta}}$. It suffices to set $\frac{\ell}{\bar{\boldsymbol{\theta}}} > \alpha$ to ensure that condition. This means we can cast *any* quadratic function that does not depends on $\boldsymbol{\theta}$ into one that depends on $\boldsymbol{\theta}$, such that $\|\partial_{\boldsymbol{\theta}} \nabla f(\boldsymbol{x}_0(\boldsymbol{\theta}), \boldsymbol{\theta})\|_F = 0$.

In such a case, the master identity from Theorem 1 reads

$$\partial \boldsymbol{x}_t(\boldsymbol{\theta}) - \partial \boldsymbol{x}_\star(\boldsymbol{\theta}) = (P_t(\boldsymbol{H}(\boldsymbol{\theta})) - \boldsymbol{H}(\boldsymbol{\theta})P'_t(\boldsymbol{H}(\boldsymbol{\theta})))(\partial \boldsymbol{x}_0(\boldsymbol{\theta}) - \partial \boldsymbol{x}_\star(\boldsymbol{\theta})),$$

where $\boldsymbol{H}(\boldsymbol{\theta}) = \boldsymbol{A}^\top \boldsymbol{A} + \boldsymbol{\theta}\boldsymbol{I}$. Now, write $Q_t(\lambda) = P_t(\lambda) - \lambda P'_t(\lambda)$. We now have the following identity,

$$\partial \boldsymbol{x}_t(\boldsymbol{\theta}) - \partial \boldsymbol{x}_\star(\boldsymbol{\theta}) = Q_t(\boldsymbol{H}(\boldsymbol{\theta}))(\partial \boldsymbol{x}_0(\boldsymbol{\theta}) - \partial \boldsymbol{x}_\star(\boldsymbol{\theta})).$$

This identity is similar to the one we have in optimization:

$$\boldsymbol{x}_t - \boldsymbol{x}_\star = P_t(\boldsymbol{H})(\boldsymbol{x}_0 - \boldsymbol{x}_\star), \qquad P_t(0) = 1,$$

and for this identity, we have the lower bound (Nemirovski, 1995, Proposition 12.3.2)

$$\|\boldsymbol{x}_t - \boldsymbol{x}_\star\|_F \geq \frac{2}{\xi^t + \xi^{-t}} \|\boldsymbol{x}_0 - \boldsymbol{x}_\star\|_F.$$

However, in the case of unrolling, we have different constraints on $Q_t$, which are the following:

$$Q_t(0) = P_t(0) - 0 \cdot P'_t(0) = 1, \qquad Q'_t(0) = P'_t(0) - P'_t(0) - 0 \cdot P''(0) = 0.$$

Therefore, we have *more* constraints on $Q$ (i.e., on how fast we can decrease the accuracy bound). Since we have seen that the functional class we work on is at least as large as the one of quadratic optimization, the lower bound can only be worse than the one for minimizing quadratic function with a bounded spectrum. $\square$

## C.5 Proof of Proposition 2

**Proposition 2.** *Assume that $\|\partial \boldsymbol{H}(\boldsymbol{\theta})(\boldsymbol{x}_0(\boldsymbol{\theta}) - \boldsymbol{x}_\star(\boldsymbol{\theta}))\|_F \leq \eta\|\partial \boldsymbol{x}_0(\boldsymbol{\theta}) - \partial \boldsymbol{x}_\star(\boldsymbol{\theta})\|_F$. Then, under Assumption 1, 2 and 3, we have the following bound for the average-case rate*

$$\mathbb{E}_{\boldsymbol{H}(\boldsymbol{\theta})}\|\partial \boldsymbol{x}_t(\boldsymbol{\theta}) - \boldsymbol{x}_\star(\boldsymbol{\theta})\|_F^2 \leq 2\|P_t\|_\eta^2 \, \mathbb{E}_{\boldsymbol{H}(\boldsymbol{\theta})}\|\partial \boldsymbol{x}_0(\boldsymbol{\theta}) - \partial \boldsymbol{x}_\star(\boldsymbol{\theta})\|_F^2.$$

*Proof.* We first derive both sides of (4) and use Assumption 2, then we use Cauchy-Schwartz and $(a+b)^2 \leq 2a^2 + 2b^2$:

$$\|\partial \boldsymbol{x}_t(\boldsymbol{\theta}) - \boldsymbol{x}_\star(\boldsymbol{\theta})\|_F^2,$$

$$= \|P_t(\boldsymbol{H}(\boldsymbol{\theta}))(\partial \boldsymbol{x}_0(\boldsymbol{\theta}) - \partial \boldsymbol{x}_\star(\boldsymbol{\theta})) + P'(\boldsymbol{H}(\boldsymbol{\theta}))\partial \boldsymbol{H}(\boldsymbol{\theta})(\boldsymbol{x}_0(\boldsymbol{\theta}) - \boldsymbol{x}_\star(\boldsymbol{\theta}))\|^2,$$

$$\leq \left( \|P_t(\boldsymbol{H}(\boldsymbol{\theta}))(\partial \boldsymbol{x}_0(\boldsymbol{\theta}) - \partial \boldsymbol{x}_\star(\boldsymbol{\theta}))\|_F + \|P'(\boldsymbol{H}(\boldsymbol{\theta}))\partial \boldsymbol{H}(\boldsymbol{\theta})(\boldsymbol{x}_0(\boldsymbol{\theta}) - \boldsymbol{x}_\star(\boldsymbol{\theta}))\|_F \right)^2,$$

$$\leq 2\|P_t(\boldsymbol{H}(\boldsymbol{\theta}))(\partial \boldsymbol{x}_0(\boldsymbol{\theta}) - \partial \boldsymbol{x}_\star(\boldsymbol{\theta}))\|_F + 2\|P'(\boldsymbol{H}(\boldsymbol{\theta}))\partial \boldsymbol{H}(\boldsymbol{\theta})(\boldsymbol{x}_0(\boldsymbol{\theta}) - \boldsymbol{x}_\star(\boldsymbol{\theta}))\|_F^2,$$

$$\leq 2\|P_t(\boldsymbol{H}(\boldsymbol{\theta}))\|_F^2\|\partial \boldsymbol{x}_0(\boldsymbol{\theta}) - \partial \boldsymbol{x}_\star(\boldsymbol{\theta})\|_F^2 + 2\|P'(\boldsymbol{H}(\boldsymbol{\theta}))\|_F^2\|\partial \boldsymbol{H}(\boldsymbol{\theta})(\boldsymbol{x}_0(\boldsymbol{\theta}) - \boldsymbol{x}_\star(\boldsymbol{\theta}))\|_F^2,$$

$$\leq 2 \left( \|P_t(\boldsymbol{H}(\boldsymbol{\theta}))\|_F^2 + \eta\|P'(\boldsymbol{H}(\boldsymbol{\theta}))\|_F^2 \right) \|\partial \boldsymbol{x}_0(\boldsymbol{\theta}) - \partial \boldsymbol{x}_\star(\boldsymbol{\theta})\|_F^2.$$

$$= 2 \left( \mathbf{Trace}(P_t(\boldsymbol{H}(\boldsymbol{\theta}))^2) + \eta\mathbf{Trace}(P'(\boldsymbol{H}(\boldsymbol{\theta}))^2) \right) \|\partial \boldsymbol{x}_0(\boldsymbol{\theta}) - \partial \boldsymbol{x}_\star(\boldsymbol{\theta})\|_F^2.$$

Since the trace of a symmetric matrix is the sum of its eigenvalues, after taking the expectation on both sides, we obtain the desired result:

$$\mathbb{E}\left[\|\partial \boldsymbol{x}_t(\boldsymbol{\theta}) - \boldsymbol{x}_\star(\boldsymbol{\theta})\|_F^2\right] \leq 2 \left( \int_\mathbb{R} P_t^2 \, \mathrm{d}\mu + \eta \int_\mathbb{R} (P_t')^2 \, \mathrm{d}\mu \right) \|\partial \boldsymbol{x}_0(\boldsymbol{\theta}) - \partial \boldsymbol{x}_\star(\boldsymbol{\theta}))\|_F^2,$$

$$\square$$

### C.6 Proof of Proposition 3

**Proposition 3.** *Let $\{S_t\}$ be a sequence of orthogonal Sobolev polynomials, i.e., $\langle S_i, S_j \rangle > 0$ if $i = j$ and $0$ otherwise, normalized such that $S_i(0) = 1$. Then, the residual polynomial that minimizes the Sobolev norm can be constructed as*

$$P_t^\star = \underset{P \in \mathcal{P}_t:P(0)=1}{\arg\min} \langle P, P \rangle_\eta = \frac{1}{A_t}\sum_{i=0}^t a_i S_i, \quad \text{where} \quad a_i = \frac{1}{\|S_t\|_\eta^2} \quad \text{and} \quad A_t = \sum_{i=0}^t a_i \, .$$

*Moreover, we have that $\|P_t^\star\|_\eta^2 = 1/A_t$.*

*Proof.* We have that the sequence $\{S_i\}_{i=0...t}$ is a orthogonal basis for $\mathcal{P}_t$. Therefore, we can write *any* polynomials as a weighted sum of $S_i$. Also, since $P_t(0) = 1$ and $S_i(0) = 1$, we have to enforce that the linear combination sums to one. This means that

$$P_t = \sum_{i=0}^t a_i S_i, \qquad \sum_{i=0}^t a_i = 1 \, .$$

We now minimize over $\alpha$.

$$\min_{P \in \mathcal{P}_t:P(0)=1} \langle P, P \rangle_\eta = \min_{\alpha: \sum_{i=0}^t a_i = 1} \langle \sum_{i=0}^t a_i S_i, \sum_{i=0}^t a_i S_i \rangle_\eta$$

$$= \min_{\alpha: \sum_{i=0}^t a_i = 1} \sum_{i=0}^t a_i^2 \langle S_i, S_i \rangle_\eta + \sum_{i=0}^t \sum_{j=0 \neq i}^t a_i \alpha_j \underbrace{\langle S_i, S_j \rangle_\eta}_{=0}$$

$$= \min_{\alpha: \sum_{i=0}^t a_i = 1} \sum_{i=0}^t a_i^2 \|S_i\|_\eta^2 \, .$$

The Lagrangian of the optimization problem reads

$$\mathcal{L}(\alpha, \lambda) = \sum_{i=0}^t a_i^2 \|S_i\|_\eta^2 + \lambda(1 - \sum_{i=0}^t a_i).$$

Taking its derivative to zero gives the desired result:

$$2a_i\|S_i\|_\eta^2 - \lambda = 0 \quad \Rightarrow \quad a_i = \frac{\lambda}{2\|S_i\|_\eta^2}, \qquad \lambda = \frac{1}{\sum_{i=0}^t a_i} \, .$$

Injecting the optimal solution into $\|P\|_\eta^2$ gives

$$\|P\|_\eta^2 = \sum_{i=0}^{t} a_i^2 \|S_i\|_\eta^2$$

$$= \left( \frac{1}{\sum_{i=0}^{t} \frac{1}{\|S_i\|_\eta^2}} \right)^2 \sum_{i=0}^{t} \frac{1}{\|S_i\|_S^4} \|S_i\|_\eta^2$$

$$= \left( \frac{1}{\sum_{i=0}^{t} \frac{1}{\|S_i\|_\eta^2}} \right)^2 \sum_{i=0}^{t} \frac{1}{\|S_i\|_\eta^2}$$

$$= \frac{1}{\sum_{i=0}^{t} \frac{1}{\|S_i\|_\eta^2}} = \frac{1}{\sum_{i=0}^{t} a_i} \ .$$

$\square$

# D  Optimal Sobolev algorithm

We recall the Sobolev algorithm:

$$\boldsymbol{y}_t = \boldsymbol{y}_{t-1} - h_t \nabla f(\boldsymbol{y}_{t-1}) + m_t(\boldsymbol{y}_{t-1} - \boldsymbol{y}_{t-2})$$
$$\boldsymbol{z}_t = c_t^{(1)} \boldsymbol{z}_{t-2} + c_t^{(2)} \boldsymbol{y}_t - c_t^{(3)} \boldsymbol{y}_{t-2}$$
$$\boldsymbol{x}_t = \frac{A_{t-1}}{A_t} \boldsymbol{x}_{t-1} + \frac{a_t}{A_t} \boldsymbol{z}_t,$$

parametrized by:

- $[\ell, L]$, lower and upper bound on the eigenvalues of $\boldsymbol{H}(\boldsymbol{\theta})$,

- $\alpha$, parameter of the Gegenbaueur distribution (12), supposed to be the expected spectral density (6). **Note:** $\alpha = 0$ leads to a sequence of Chebyshev polynomials for $y_t$.

- $\eta$, assumed to satisfy the inequality $\|\partial \boldsymbol{H}(\boldsymbol{\theta})(\boldsymbol{x}_0(\boldsymbol{\theta}) - \boldsymbol{x}_\star(\boldsymbol{\theta}))\|_F \leq \eta \|\partial \boldsymbol{x}_0(\boldsymbol{\theta}) - \partial \boldsymbol{x}_\star(\boldsymbol{\theta})\|_F$. Intuitively, this parameter is the balance between $\|P\|$ and $\|P'\|$.

## D.1  Initialization (required for $t = 0$ and $t = 1$)

### D.1.1  Side parameters

$$y_0 = z_0 = x_0$$
$$\delta_1 = -\frac{L - \ell}{L + \ell}$$
$$\kappa_1 = 1$$
$$\kappa_2 = 1$$
$$d_0 = \xi_0$$
$$d_1 = \frac{3}{2(\alpha + 2)(\alpha + 1)(1 + 2\eta(\alpha + 1))},$$
$$d_2 = \frac{3}{(\alpha + 3)(\alpha + 2)\left(1 + \eta\frac{8(\alpha + 2)(\alpha + 1)}{2\alpha + 1}\right)},$$

### D.1.2  Main parameters

$$h_1 = -\frac{2\delta_1}{L - \ell}$$
$$m_1 = -\left(1 + \delta_1 \frac{L + \ell}{L - \ell}\right)$$
$$c_1^{(1)} = 0$$
$$c_1^{(2)} = 1$$
$$c_1^{(3)} = 0$$
$$a_1 = \frac{d_1}{\xi_1 K_1}\left(\frac{L + \ell}{L - \ell}\right)^2,$$
$$A_1 = A_0 + a_1$$

## D.2 Recurrence (for $t \geq 2$)

### D.2.1 Side parameters

$$\gamma_t = \frac{t(t + 2\alpha - 1)}{4(t + \alpha)(t + \alpha + 1)}$$

$$\delta_t = \frac{1}{-\frac{L+\ell}{L-\ell} + \delta_{t-1}\gamma_t}$$

$$\xi_t = \frac{(t + 2)(t + 1)}{4(t + \alpha + 1)(t + \alpha)}$$

$$d_t = \frac{\xi_t \gamma_t \gamma_{t-1}}{\gamma_{t-1}(\eta t^2 + \gamma_t) + \xi_{t-2}(\xi_{t-2} - d_{t-2})}$$

$$\Delta_t^P = \frac{1 + \delta_t \frac{L+\ell}{L-\ell}}{\gamma_t},$$

$$\kappa_t = \frac{1}{1 + \left(\frac{d_{t-2}}{\kappa_{t-2}} - \xi_{t-2}\right)\Delta_t^P},$$

$$\tau_t = \frac{1}{\frac{d_{t-2}}{\kappa_{t-2}} + \frac{1}{\Delta_t^P} - \xi_{t-2}},$$

$$\Delta_t^S = \frac{1}{d_{t-2} + \left(\frac{1}{\Delta_t^P} - \xi_{t-2}\right)\kappa_{t-2}}$$

$$K_t = \frac{t(t - 1 + 2\alpha)}{4(t + \alpha - 1)(t + \alpha)},$$

### D.2.2 Main parameters

$$h_t = -\frac{2\delta_t}{L - \ell}$$

$$m_t = -\left(1 + \delta_t \frac{L + \ell}{L - \ell}\right)$$

$$c_t^{(1)} = d_{t-2}\Delta_t^S$$

$$c_t^{(2)} = \kappa_t$$

$$c_t^{(3)} = -\tau_t \xi_{t-2}$$

$$a_t = \frac{d_t \xi_{t-2}}{\xi_i d_{t-2} K_t K_{t-1} \Delta_i^2} a_{t-2},$$

$$A_t = A_{t-1} + a_t$$

# E   Derivation of the Sobolev algorithm

## E.1   Notations

In this section, we use the following notations. We denote by $\mu$ the Gegenbaueur density 12 defined in $[\ell, L]$, $\tilde{\mu}$ the Gegenbaueur density defined in $[-1, 1]$:

$$\mu(\lambda) = \tilde{\mu}(m(\lambda)), \quad \tilde{\mu}(x) = (1 - x^2)^{\alpha - \frac{1}{2}} \quad \text{and} \quad m : [\ell, L] \to [0, 1], \, m(\lambda) = \frac{2\lambda - L - \ell}{L - \ell}.$$

where

$$m(\lambda) = \underbrace{\frac{2}{L - \ell}}_{= \sigma_1} \lambda + \underbrace{\left(-\frac{L + \ell}{L - \ell}\right)}_{= \sigma_0} \tag{19}$$

We also denote by $G_t$ and $\tilde{G}_t$ the sequence of Gegenbaueur polynomials that are orthogonal respectively w.r.t. the measure $\mu$ and $\tilde{\mu}$, that it, for all $i, j \geq 0$, we have

$$\int_\mu^L G_i(\lambda) G_j(\lambda) \, d\mu(\lambda) \begin{cases} > 0 & \text{if } i = j \\ = 0 & \text{otherwise} \end{cases} \qquad \int_{-1}^1 \tilde{G}_i(x) \tilde{G}_j(x) \, d\mu(x) \begin{cases} > 0 & \text{if } i = j \\ = 0 & \text{otherwise} \end{cases}$$

In terms of normalization, we have that $G_t$ is a *residual* polynomial, and $\tilde{G}_t$ is a *monic* polynomials. In other terms,

$$G(\lambda) = \mathbf{1} + ... \lambda^1 + ... + ... \lambda^t, \qquad \tilde{G}(\lambda) = ... x^0 + ... x^1 + ... + \mathbf{1} x^t$$

In such a case, by using the linear mapping $m(\lambda)$ from $[\ell, L]$ to $[-1, 1]$, see (19), we have the following relation:

$$G_t(\lambda) = \frac{\tilde{G}_t(m(\lambda))}{\tilde{G}_t(m(0))}. \tag{20}$$

Similarly, we define $S_t$ and $\tilde{S}_t$ the sequence of orthogonal Sobolev polynomials w.r.t. the Sobolev product involving the Gegenbaueur density, i.e.,

$$\int_\mu^L S_i(\lambda) S_j(\lambda) \, d\mu(\lambda) + \eta \int_\mu^L S_i'(\lambda) S_j'(\lambda) \, d\mu(\lambda) \begin{cases} > 0 & \text{if } i = j \\ = 0 & \text{otherwise} \end{cases},$$

and

$$\int_{-1}^1 \tilde{S}_i(x) \tilde{S}_j(x) \, d\mu(x) + \tilde{\eta} \int_{-1}^1 \tilde{S}_i'(x) \tilde{S}_j'(x) \, d\mu(x) \begin{cases} > 0 & \text{if } i = j \\ = 0 & \text{otherwise} \end{cases}.$$

Originally, they are called Gegenbaueur-Sobolev polynomials (Marcellán et al., 1994) because $\mu$ is a Gegenbaueur density, but for conciseness, we simply call them Sobolev polynomials. As for the Gegenbaueur polynomials, $S_t$ is a *residual* polynomial while $\tilde{S}_t$ is a *monic* polynomial. Finally, we have that

$$S_t(\lambda) = \frac{\tilde{S}_t(m(\lambda))}{\tilde{S}_t(m(0))} \quad \text{if and only if} \quad \tilde{\eta} = \sigma_1^2 \eta. \tag{21}$$

Note that we make a distinction between plain symbols and tilde ˜ symbols, where the tilde ˜ notation is used for polynomials that are defined on $[-1, 1]$, while the plain notation is the counterpart defined on $[\ell, L]$.

## E.2   Monic Sobolev polynomial

We now describe the construction of $\tilde{S}$, detailed in (Marcellán et al., 1994). The monic Gegenbaueur polynomial is constructed as

$$\tilde{G}_0 = 1, \quad \tilde{G}_1 = x, \quad \tilde{G}_{t+1}(x) = x\tilde{G}_t(x) - \gamma_t \tilde{G}_{t-1}(x), \quad \gamma_t = \frac{t(t + 2\alpha + 1)}{4(t + \alpha)(t + \alpha - 1)}. \tag{22}$$

Then, the Sobolev polynomials are defined as a simple recurrence involving $\tilde{G}_t$ and $\tilde{G}_{t-2}$,

$$\tilde{S}_0 = \tilde{G}_0, \quad \tilde{S}_1 = \tilde{G}_1, \quad \tilde{S}_t = d_{t-2} \tilde{S}_{t-2} + \tilde{G}_t - \xi_{t-2} \tilde{G}_{t-2}, \tag{23}$$

where

$$\xi_t = \frac{(t+2)(t+1)}{4(t+\alpha+1)(t+\alpha)},$$

$$d_0 = \xi_0,$$

$$d_1 = \frac{3}{2(\alpha+2)(\alpha+1)(1+2\eta(\alpha+1))},$$

$$d_2 = \frac{3}{(\alpha+3)(\alpha+2)\left(1+\eta\frac{8(\alpha+2)(\alpha+1)}{2\alpha+1}\right)},$$

$$d_t = \frac{\xi_t\gamma_t\gamma_{t-1}}{\gamma_{t-1}(\eta t^2+\gamma_t)+\xi_{t-2}(\xi_{t-2}-d_{t-2})}.$$

(24)

Note that the following property will be important later:

$$d_t = \xi_t \frac{\|\tilde{G}_t\|}{\|\tilde{S}_t\|_{\tilde{\eta}}},$$

(25)

where

$$\|\tilde{G}_t\|^2 = \int_{-1}^{1} \tilde{G}_t^2(x)\,\mathrm{d}\mu(x), \qquad \|\tilde{S}_t\|_{\tilde{\eta}} = \int_{-1}^{1} \tilde{S}_t^2(x)\,\mathrm{d}\mu(x) + \tilde{\eta}\int_{-1}^{1}[\tilde{S}_t'(x)]^2\,\mathrm{d}\mu(x).$$

### E.3  Shifted, normalized Sobolev polynomials

We now shift and normalize the Sobolev polynomials, that it, instead of being defined in $[0,1]$ and being monic, we make them defined in $[\ell, L]$ (evaluate the polynomial at $x = m(\lambda)$) and residual (divide the polynomial by $\tilde{S}_t(m(0))$).

We begin by doing it to the Gegenbaueur polynomials. By applying the technique from (Pedregosa et al., 2020, Proposition 18) on the polynomial $\tilde{G}_t(m(\lambda))$,

$$\tilde{G}_t(m(\lambda)) = \sigma_0\tilde{G}_{t-1}(m(\lambda)) + \sigma_1\lambda\tilde{G}_{t-1}(m(\lambda)) - \gamma_{t-1}\tilde{G}_{t-2}(m(\lambda)).$$

We obtain the recurrence

$$G_t(m(\lambda)) = \sigma_0\delta_t G_{t-1}(m(\lambda)) + \sigma_1\delta_t\lambda G_{t-1}(m(\lambda)) + (1-\sigma_0\delta_t)\tilde{G}_{t-2}(m(\lambda)),$$

(26)

where

$$\delta_t = \frac{\tilde{G}_{t-1}(m(0))}{\tilde{G}_t(m(0))} = \frac{1}{\sigma_0 - \delta_{t-1}\gamma_{t-1}}.$$

(27)

This expression can be cast into a recurrence that involves a step size and a momentum,

$$G_t(\lambda) = G_{t-1} - h_t\lambda G_{t-1}(\lambda) + m_t(G_{t-1}(\lambda) - G_{t-2}(\lambda)),$$

where

$$\delta_1 = -\frac{L-\ell}{L+\ell},$$

$$h_1 = -\frac{2\delta_1}{L-\ell},$$

$$m_1 = -\left(1+\delta_1\frac{L+\ell}{L-\ell}\right),$$

$$\delta_t = \frac{1}{-\frac{L+\ell}{L-\ell}+\delta_{t-1}\gamma_{t-1}},$$

$$h_t = -\frac{2\delta_t}{L-\ell},$$

$$m_t = -\left(1+\delta_t\frac{L+\ell}{L-\ell}\right).$$

We now show how to shift and normalize the Sobolev polynomial. The shifting operation is not complicated, as it suffice to evaluate the polynomial $\tilde{S}_t$ at $x = m(\lambda)$. The difficult part is the normalization. Using the relations (20), (21) and (23), we obtain

$$S_t = \underbrace{\frac{\tilde{S}_{t-2}(m(0))}{\tilde{S}_t(m(0))} d_{t-2}}_{=c_t^{(1)}} S_{t-2} + \underbrace{\frac{\tilde{G}_t(m(0))}{\tilde{S}_t(m(0))}}_{=c_t^{(2)}} G_t - \underbrace{\frac{\tilde{G}_{t-2}(m(0))}{\tilde{S}_t(m(0))}}_{=c_t^{(3)}} G_{t-2}.$$

Therefore, we have to compute those quantities that involves ratio of polynomials evaluated at $\lambda = 0$, whose recurrence is detailed in the next Proposition.

**Proposition 4.** *Let*

$$\Delta_t^P = \frac{\tilde{G}_{t-2}(m(0))}{\tilde{G}_t(m(0))}, \qquad \Delta_t^S = \frac{\tilde{S}_{t-2}(m(0))}{\tilde{S}_t(m(0))} \qquad \kappa_t = \frac{\tilde{G}_t(m(0))}{\tilde{S}_t(m(0))}, \qquad \tau_t = \frac{\tilde{G}_{t-2}(m(0))}{\tilde{S}_t(m(0))}.$$

*Then,*

$$\Delta_t^P = \delta_t \delta_{t-1} = \frac{\sigma_0 \delta_t - 1}{\gamma_{t-1}}, \tag{28}$$

$$\kappa_t = \frac{1}{1 + \left(\frac{d_{t-2}}{\kappa_{t-2}} - \xi_{t-2}\right) \Delta_t^P}, \tag{29}$$

$$\tau_t = \frac{1}{\frac{d_{t-2}}{\kappa_{t-2}} + \frac{1}{\Delta_t^P} - \xi_{t-2}}, \tag{30}$$

$$\Delta_t^S = \frac{1}{d_{t-2} + \left(\frac{1}{\Delta_t^P} - \xi_{t-2}\right) \kappa_{t-2}} \tag{31}$$

*Proof.* We now show, one by one, each terms of the recurrence. We begin by $\Delta_t^P$. Indeed,

$$\tilde{G}_t(m(\lambda)) = \sigma_0 \tilde{G}_{t-1}(m(\lambda)) + \sigma_1 m(\lambda) \tilde{G}_{t-1}(m(\lambda)) - \gamma_{t-1} \tilde{G}_{t-2}(m(\lambda)).$$

Therefore, using (27), we obtain

$$G_t(m(\lambda)) = \sigma_0 \delta_t \tilde{G}_{t-1}(m(\lambda)) + \sigma_1 \delta_t m(\lambda) \tilde{G}_{t-1}(m(\lambda)) - \gamma_{t-1} \Delta_t^P \tilde{G}_{t-2}(m(\lambda)).$$

After comparing this expression with (26), we deduce that

$$-\gamma_{t-1} \Delta_t^P = (1 - \sigma_0 \delta_t).$$

In other words,

$$\Delta_t^P = \frac{\sigma_0 \delta_t - 1}{\gamma_{t-1}}.$$

To show the other recurrences, we will often use the fact that

$$\tilde{S}_t(m(0)) = d_{t-2} \tilde{S}_{t-2}(m(0)) + \tilde{G}_t(m(0)) - \xi_{t-2} \tilde{G}_{t-2}(m(0)). \tag{32}$$

We now show how to form $\tau_t$. Indeed, using (32),

$$\begin{aligned}
\tau_t^{-1} &= \frac{\tilde{S}_t(m(0))}{\tilde{G}_{t-2}(m(0))} \\
&= \frac{d_{t-2} \tilde{S}_{t-2}(m(0)) + \tilde{G}_t(m(0)) - \xi_{t-2} \tilde{G}_{t-2}(m(0))}{\tilde{G}_{t-2}(m(0))} \\
&= \frac{d_{t-2}}{\kappa_{t-2}} + \frac{1}{\Delta_t^P} - \xi_{t-2}.
\end{aligned}$$

Using the same technique, we have for $\kappa_t$:

$$\kappa_t^{-1} = \frac{\tilde{S}_t(m(0))}{\tilde{G}_t(m(0))}$$

$$= \frac{d_{t-2}\tilde{S}_{t-2}(m(0)) + \tilde{G}_t(m(0)) - \xi_{t-2}\tilde{G}_{t-2}(m(0))}{\tilde{G}_t(m(0))}$$

$$= d_{t-2}\frac{\tilde{S}_{t-2}(m(0))}{\tilde{G}_t(m(0))} + 1 - \xi_{t-2}\Delta_t^P$$

However,

$$\frac{\tilde{S}_{t-2}(m(0))}{\tilde{G}_t(m(0))} = \frac{\tilde{S}_{t-2}(m(0))}{\tilde{G}_{t-2}(m(0))}\frac{\tilde{G}_{t-2}(m(0))}{\tilde{G}_t(m(0))} = \frac{\Delta_t^P}{\kappa_{t-2}}.$$

Therefore,

$$\kappa_t^{-1} = d_{t-2}\frac{\Delta_t^P}{\kappa_{t-2}} + 1 - \xi_{t-2}\Delta_t^P = 1 + \left(\frac{d_{t-2}}{\kappa_{t-2}} - \xi_{t-2}\right)\Delta_t^P$$

Finally, it remains to show the recurrence for $\Delta_t^S$. As usual,

$$(\Delta_t^S)^{-1} = \frac{\tilde{S}_t(m(0))}{\tilde{S}_{t-2}(m(0))}$$

$$= \frac{d_{t-2}\tilde{S}_{t-2}(m(0)) + \tilde{G}_t(m(0)) - \xi_{t-2}\tilde{G}_{t-2}(m(0))}{\tilde{S}_{t-2}(m(0))}$$

$$= d_{t-2} + \frac{\tilde{G}_t(m(0))}{\tilde{S}_{t-2}(m(0))} - \xi_{t-2}\kappa_{t-2}$$

We have seen before that

$$\frac{\tilde{S}_{t-2}(m(0))}{\tilde{G}_t(m(0))} = \frac{\Delta_t^P}{\kappa_{t-2}},$$

which finally gives

$$(\Delta_t^S)^{-1} = d_{t-2} + \frac{\kappa_{t-2}}{\Delta_t^P} - \xi_{t-2}\kappa_{t-2} = d_{t-2} + \left(\frac{1}{\Delta_t^P} - \xi_{t-2}\right)\kappa_{t-2}.$$

$\square$

### E.4 Norm of Sobolev Polynomials

Now that we can build the shifted, normalized Gegenbaueur and Sobolev polynomials, we still need to compute the norm of the Sobolev polynomial to compute $P_t^\star$.

First, for simplicity, we write

$$\|G_t\|^2 = \int_\ell^L G_t^2(\lambda)\,\mathrm{d}\mu(\lambda)$$

$$\|\tilde{G}_t\|^2 = \int_{-1}^1 \tilde{G}_t^2(x)\,\mathrm{d}\tilde{\mu}(x)$$

$$\|S_t\|_\eta^2 = \int_\ell^L S_t^2(\lambda) + \eta[S_t'(\lambda)]^2\,\mathrm{d}\mu(\lambda)$$

$$\|\tilde{S}_t\|_{\tilde{\eta}}^2 = \int_{-1}^1 \tilde{S}_t^2(x) + \tilde{\eta}[\tilde{S}_t'(x)]^2\,\mathrm{d}\tilde{\mu}(x), \quad \tilde{\eta} = \sigma_1^2\eta$$

Indeed, to obtain the optimal method, we need to compute the coefficients

$$a_t = \frac{1}{\|S_t\|_\eta^2}.$$

To do so, we will use the property (25):

$$d_t = \xi_t \frac{\|\tilde{G}_t\|^2}{\|\tilde{S}_t\|_{\tilde{\eta}}^2}.$$

We begin by the explicit expression of the norm of the shifted, normalized Sobolev polynomials, and express it as a function of the norm of the plain, monic Sobolev polynomial. Indeed,

$$\|S_t(\lambda)\|_\eta^2 = \int_\ell^L \frac{\tilde{S}_t^2(m(\lambda))}{\tilde{S}_t^2(m(0))} + \eta \frac{[m'(\lambda)\tilde{S}_t'(m(\lambda))]^2}{\tilde{S}_t^2(m(0))} \, d\mu(\lambda)$$

Since $m'(\lambda) = \sigma_1$, and since $\tilde{\eta} = \sigma_1 \eta$, we have

$$\begin{aligned}
\|S_t(\lambda)\|_\eta^2 &= \frac{1}{\tilde{S}_t^2(m(0))} \int_\ell^L \tilde{S}_t^2(m(\lambda)) + \tilde{\eta}[\tilde{S}_t'(m(\lambda))]^2 \, d\mu(\lambda) \\
&= \frac{1}{\tilde{S}_t^2(m(0))} \int_\ell^L \tilde{S}_t^2(m(\lambda)) + \tilde{\eta}[\tilde{S}_t'(m(\lambda))]^2 \, d\tilde{\mu}(m(\lambda)) \\
&= \frac{1}{\tilde{S}_t^2(m(0))} \int_{-1}^1 \left( \tilde{S}_t^2(x) + \tilde{\eta}[\tilde{S}_t'(x)]^2 \, d \right) \frac{\tilde{\mu}(x)}{m'(x)} \\
&= \frac{\sigma_1}{\tilde{S}_t^2(m(0))} \int_{-1}^1 \left( \tilde{S}_t^2(x) + \tilde{\eta}[\tilde{S}_t'(x)]^2 \, d \right) \tilde{\mu}(x) \\
&= \frac{\sigma_1}{\tilde{S}_t^2(m(0))} \|\tilde{S}_t\|_{\tilde{\eta}}^2 
\end{aligned} \tag{33}$$

Note that, by definition of $\Delta_t^S$, we have the recursion

$$\tilde{S}_t^2(m(0)) = \frac{\tilde{S}_{t-2}^2(m(0))}{[\Delta_t^S]^2}. \tag{34}$$

Let $\bar{G}_t$ be defined as

$$\bar{Q}_t = \frac{1}{t} \left[ 2x(t+\alpha-1)\bar{Q}_{t-1} - (t+2\alpha-2)\bar{Q}_{t-2} \right],$$

i.e., $\bar{G}_t$ is a scaled version of $G_t$, which is the classical definition of Gegenbaueur polynomials. Then

$$\|\bar{G}_t\|^2 = \frac{\pi 2^{(1-2\alpha)}}{[\Gamma(\alpha)]^2} \frac{\Gamma(t+2\alpha)}{t!(t+\alpha)}.$$

Since $\Gamma(x+1) = x\Gamma(x)$, we can deduce a recurrence equation. Indeed,

$$\begin{aligned}
\|\bar{G}_t\|^2 &= \frac{\pi 2^{(1-2\alpha)}}{[\Gamma(\alpha)]^2} \frac{\Gamma(t+2\alpha)}{t!(t+\alpha)} \\
&= \frac{\pi 2^{(1-2\alpha)}}{[\Gamma(\alpha)]^2} \frac{(t-1+2\alpha)\Gamma(t-1+2\alpha)}{t(t-1)!(t+\alpha)} \\
&= \frac{\pi 2^{(1-2\alpha)}}{[\Gamma(\alpha)]^2} \frac{(t-1+2\alpha)}{t} \frac{t-1+\alpha}{t+\alpha} \frac{\Gamma(t-1+2\alpha)}{(t-1)!(t-1+\alpha)} \\
&= \frac{(t-1+2\alpha)(t-1+\alpha)}{t(t+\alpha)} \|\bar{G}_{t-1}\|^2.
\end{aligned} \tag{35}$$

with the initial condition

$$\|\bar{G}_0\|^2 = \frac{\pi 2^{(1-2\alpha)}}{[\Gamma(\alpha)]^2} \frac{\Gamma(2\alpha)}{0!\alpha} = \frac{\pi 2^{(1-2\alpha)}\Gamma(2\alpha)}{\alpha[\Gamma(\alpha)]^2}.$$

However, there is a factor between $\bar{G}_t$ and the monic polynomial $\tilde{G}_t$. Indeed,

$$\tilde{G}_t = \frac{\bar{G}_t}{\prod_{i=0}^t \frac{2(i+\alpha-1)}{i}}. \tag{36}$$

This factor can be computed recursively. Let $k_t = \frac{1}{\prod_{i=0}^{t} \frac{2(i+\alpha-1)}{i}}$. Then,

$$
\begin{aligned}
k_t &= \prod_{i=0}^{t} \frac{i}{2(i+\alpha-1)} \\
&= \frac{t}{2(t+\alpha-1)} \prod_{i=0}^{t-1} \frac{i}{2(i+\alpha-1)} \\
&= \frac{t}{2(t+\alpha-1)} k_{t-1}.
\end{aligned}
\tag{37}
$$

Therefore, using successively (35), (37), then (36), we have

$$
\begin{aligned}
\|\tilde{G}_t\|^2 &= k_t^2 \|\bar{G}_t\|^2 \\
&= \frac{t^2}{4(t+\alpha-1)^2} k_{t-1}^2 \|\bar{G}_t\|^2 \\
&= \frac{t^2}{4(t+\alpha-1)^2} \frac{(t-1+2\alpha)(t-1+\alpha)}{t(t+\alpha)} k_{t-1}^2 \|\bar{G}_{t-1}\|^2 \\
&= \frac{t}{4(t+\alpha-1)} \frac{(t-1+2\alpha)}{(t+\alpha)} k_{t-1}^2 \|\bar{G}_{t-1}\|^2 \\
&= \underbrace{\frac{t(t-1+2\alpha)}{4(t+\alpha-1)(t+\alpha)}}_{=K_t} \|\tilde{G}_{t-1}\|^2,
\end{aligned}
\tag{38}
$$

with the same initial condition

$$
\|\bar{G}_0\|^2 = \|\tilde{G}_0\|^2 = \frac{\pi 2^{(1-2\alpha)} \Gamma(2\alpha)}{\alpha [\Gamma(\alpha)]^2}.
$$

We now compute the recursion for $\|S_t\|_\eta^2$. Indeed, by using successively (33), (34), (25), (38)$\times$2, (25) then (33),

$$
\begin{aligned}
\|S_t\|_\eta^2 &= \frac{\sigma_1}{\tilde{S}_t^2(m(0))} \|\tilde{S}_t\|_{\tilde{\eta}}^2 \\
&= \frac{\sigma_1 [\Delta_t^S]^2}{\tilde{S}_{t-2}^2(m(0))} \|\tilde{S}_t\|_{\tilde{\eta}}^2 \\
&= [\Delta_t^S]^2 \frac{\sigma_1}{\tilde{S}_{t-2}^2(m(0))} \frac{\xi_t \|\tilde{G}_t\|^2}{d_t} \\
&= [\Delta_t^S]^2 \frac{\sigma_1}{\tilde{S}_{t-2}^2(m(0))} K_t K_{t-1} \frac{\xi_t \|\tilde{G}_{t-2}\|^2}{d_t} \\
&= [\Delta_t^S]^2 \frac{\sigma_1}{\tilde{S}_{t-2}^2(m(0))} K_t K_{t-1} \frac{\xi_t d_{t-2}}{d_t \xi_{t-2}} \frac{\xi_{t-2} \|\tilde{G}_{t-2}\|^2}{d_{t-2}} \\
&= [\Delta_t^S]^2 \frac{\sigma_1}{\tilde{S}_{t-2}^2(m(0))} K_t K_{t-1} \frac{\xi_t d_{t-2}}{d_t \xi_{t-2}} \|\tilde{S}_{t-2}\|_{\tilde{\eta}}^2 \\
&= [\Delta_t^S]^2 K_t K_{t-1} \frac{\xi_t d_{t-2}}{d_t \xi_{t-2}} \|S_{t-2}\|_\eta^2.
\end{aligned}
$$

We finally have the desired recurrence for the $a_t$'s since

$$
a_i = \frac{\bar{a}}{\|S_t\|_\eta^2},
$$

where $\bar{a}$ is a nonzero multiplicative constant. We can arbitrarily decide that $\bar{a} = 1$, which gives us $a_0 = 1$. Given that $S_1 = G_1$, and after using (38), (25) and (33), we have

$$
a_1 = \frac{d_1}{\xi_1 K_1} \left( \frac{L+\ell}{L-\ell} \right)^2.
$$

# F Asymptotic algorithm

## F.1 Asymptotics of Sobolev-Gegenbaeur polynomials

From (Scieur et al., 2020b), we know that the parameters converges asymptotically to

$$h_t \to h = \left(\frac{2}{\sqrt{L}+\sqrt{\ell}}\right)^2, \quad m_t \to m = \left(\frac{\sqrt{L}-\sqrt{\ell}}{\sqrt{L}+\sqrt{\ell}}\right)^2, \quad \delta_t^P \to 2\sqrt{m}, \quad \delta_t^P \to 4m.$$

In addition, it is easy to see that

$$\xi_\infty = \frac{1}{4}, \qquad \gamma_\infty = \frac{1}{4}.$$

Therefore,

$$d_\infty = \lim_{t\to\infty} \frac{\xi_t \gamma_t \gamma_{t-1}}{\gamma_{t-1}(\eta t^2 + \gamma_t) + \xi_{t-2}(\xi_{t-2} - d_{t-2})} = \frac{\frac{1}{16}}{\eta t^2 + \frac{1}{2} - d_\infty} = O(1/t^2) \to 0.$$

Thus, the recurrence simplifies into (after replacing $d_\infty$ by 0)

$$\kappa_\infty = \frac{1}{1 - \xi_\infty \Delta_\infty^P} = \frac{1}{1-m}, \tag{39}$$

$$\tau_\infty = \frac{1}{\frac{1}{\Delta_\infty^P} - \xi_\infty} = \frac{4m}{1-m}, \tag{40}$$

$$\Delta_\infty^S = \frac{1}{\left(\frac{1}{\Delta_\infty^P} - \xi_\infty\right)\kappa_\infty} = 4m \tag{41}$$

This means that the asymptotic recurrence for $S$ reads

$$S_t = d_{t-2}\Delta_t^S S_{t-2} + \kappa_t G_t - \tau_t \xi_{t-2} G_{t-2} \to \frac{G_t - mG_{t-2}}{1-m}.$$

Moreover, we have

$$a_t = \frac{d_i \xi_{i-2}}{\xi_i d_{i-2} K_i K_{i-1} \Delta_i^2} a_{t-2},$$

$$K_t = \frac{t(t-1+2\alpha)}{4(t+\alpha-1)(t+\alpha)},$$

$$a_0 = 1$$

$$a_1 = \frac{d_1 \sigma_0^2}{\xi_1 K_1}$$

When $t \to \infty$, we have that $K_t \to 1/4$, $\xi_t \to 1/4$, $\Delta_t \to 4m$. Therefore,

$$\lim_{t\to\infty} \frac{a_t}{a_{t-2,}} = \lim_{t\to\infty} \frac{d_t}{d_{t-2}m^2}$$

Moreover, $\frac{d_i}{d_{i-2}} \to 1$. So, we have in the end that

$$\lim_{t\to\infty} \frac{a_t}{a_{t-2,}} = \frac{1}{m^2},$$

or more simply,

$$\lim_{t\to\infty} \frac{a_t}{a_{t-1,}} = \frac{1}{m}.$$

Therefore, when $t \to \infty$, we have

$$\lim_{t\to\infty} \frac{A_t}{a_t} = \lim_{t\to\infty} \sum_0^t m^t = \frac{1}{1-m}. \tag{42}$$

This means that the asymptotic dynamic for $P^\star$ reads

$$P_t^\star = \frac{A_{t-1}}{A_t} P_{t-1} + \frac{a_t}{A_t} S_t \to mP_{t-1} + (1-m)S_t.$$

# G   Asymptotic algorithm and asymptotic rate

The asymptotic recurrence of the polynomials reads

$$G_t = (1+m)G_{t-1} + h\nabla x G_{t-1} - mG_{t-2},$$
$$S_t = \frac{G_t - mG_{t-2}}{1-m},$$
$$P_t^\star = mP_{t-1}^\star + (1-m)S_t.$$

This can be simplified into

$$G_t = (1+m)G_{t-1} + h\nabla x G_{t-1} - mG_{t-2},$$
$$P_t^\star = G_t + m(P_{t-1}^\star - G_{t-2}).$$

Translated into an algorithm, we finally have a weighted average of HB iterates:

$$y_t = y_{t-1} + h\nabla f(y_{t-1}) + m(y_{t-1} - y_{t-2}), \tag{43}$$
$$x_t = y_t + m(x_{t-1} - y_{t-2}). \tag{44}$$

Note that the asymptotic rate reads

$$\lim_{t\to\infty} \frac{\|P_t^\star\|}{\|P_{t-1}^\star\|} = \frac{A_{t-1}}{A_t} = m.$$

Therefore, when $t \to \infty$,

$$\|\partial x_t(\theta) - \partial x^\star(\theta)\|_F^2 \leq O(m^t \|\partial x_0(\theta) - \partial x^\star(\theta)\|_F^2).$$