# OpenReview forum: "The Curse of Unrolling: Rate of Differentiating Through Optimization"
_NeurIPS.cc/2022/Conference — NeurIPS 2022 Accept_

### Official Review · Reviewer_eLXs · 2022-07-08

**Rating:** 6
**Confidence:** 3
**Soundness:** 3 good
**Presentation:** 3 good
**Contribution:** 3 good

**Summary:**

This paper studies the use of unrolling as a way to differentiate through an optimization problem. Specifically, it focuses on the estimation of the Jacobian (of the solution w.r.t. the trainable parameters of the network) when the problem is a quadratic optimization and the algorithm is a first-order method. The main contribution is the analysis which shows ``the curse of unrolling'' and on top of it, this paper proposes a scheme to accelerate the unrolled differentiation based on Sobolev orthogonal polynomials. The experiments show curves that match well with the proposed analysis.

**Questions:**

Please see weaknesses.

**Limitations:**

The authors have discussed the limitations in detail and claim to have no potential negative societal impact.

**Strengths And Weaknesses:**

Strengths:
1. The topic is both theoretically and practically important considering that the unrolling scheme has been utilized by a large family of deep learning works.
2. The results (theorem1-3) are novel and can explain the practical observation. Discussions are given to explain how the results can lead to certain pattern in the plot.
3. The proposed accelerated scheme for unrolling has a limited ``burn-in'' phase which is a useful property.

Weaknesses:
1. My largest concern is that for deep pipelines, both the Jacobian $\partial x_t - \partial x_*$ and the final solution $||x_t-x_*||$ matter at the same time. With only the bound for the former one, the results cannot be directly used to guide the practical use of unrolling. Especially for the proposed accelerated scheme, it is not clear how this modified scheme will influence $||x_t-x_*||$ .
2. The datasets used in the experiment are kind of toy-size by modern standards. Since the topic is the use of unrolling in deep learning pipelines, it would be more convincing if more serious experiments were included.

---

> ### Author Response · Authors · 2022-08-02
> **Response to reviewer**
>
> We thank the reviewer for the time spent on the paper and their insightful questions. We hope we answered all the reviewer's concerns, especially about the optimality of the iterates of our method. We also ran additional experiments, as requested by the reviewer.
>
> ---
>
> ***Q:** My largest concern is that for deep pipelines, both the Jacobian and the final solution matter at the same time. With only the bound for the former one, the results cannot be directly used to guide the practical use of unrolling. Especially for the proposed accelerated scheme, it is not clear how this modified scheme will influence*
>
> **A:** This is a very interesting point. Thank you for bringing it out. To our knowledge, there is a large amount of deep learning applications for which the final solution of the inner problem does *not* matter. For instance, in the case of Generative Adversarial Networks [Goodfellow et al. 2014], the discriminator is used only to provide a gradient signal to the generator. At the end of the training, only the generator is kept, and the discriminator is discarded (thus, we do not care about the final solution to the inner problem). Another example is adversarial training [Madry et al. 2018], for which, at each time-step adversarial examples are computed to robustly train a classifier against these adversarial examples. At the end of the training, only the (potentially) robust classifier is kept, and the actual efficiency of the adversarial examples used for the training is not considered (the task was only to train a robust classifier).
>
> We are very curious to know what the reviewer had in mind when referring to "deep pipelines" for which "both the Jacobian and the final solution matter." Could you be more specific?
>
> That being said, let us now assume that having a good final solution is important for our deep learning application of interest. Fortunately, by design, our method also generates iterates that are optimal w.r.t. the distance to the optimum in the case of quadratic functions. Similar variants can be derived for optimality w.r.t. gradient norm or function suboptimality gap.
>
> More precisely, the iterates $y_t$ in equation (12) are generated by Gegenbaueur polynomials: This is the optimal scheme in the average case for minimizing $\|y_t-x_\star\|$ ---the scheme associated with the $P_t$ that minimizes (6)--- when the quadratic function satisfies the e.s.d. assumption described in Section 2.2.3. We can obtain the optimal worst-case scheme using the Chebyshev distribution, i.e., when setting $\alpha=0$ in equation (11).
>
> Therefore, as practical use for unrolling, we can track the optimality of the iterates $y_{t}$. Indeed, those iterates are directly correlated to the optimality w.r.t. the Jacobian accuracy of the iterates $x_t$.
>
> ---
>
> ***Q:** it would be more convincing if more serious experiments were included*
>
> **A:** We have added an experiment illustrating the curse of unrolling on a problem with a logistic regression loss in the Appendices (Experiments on Logistic regression). The figure shows the same pattern as Figure 2 for logistic regression.
>
> The only significant difference with the least squares loss is the range of step-size values that exhibit the initial burn-in phase. While for the quadratic loss, these are step-sizes close to $h=\frac{2}{L + \mu}$, in the case of logistic regression, $L$ is a crude global upper bound, and so this step-size is not necessarily the one that achieves the fastest convergence rate. The featured two-phase curve was computed using the step size with the fastest asymptotic rate, calculated through a grid search on the step-size values.

---

### Official Review · Reviewer_ZGFf · 2022-07-08

**Rating:** 6
**Confidence:** 3
**Soundness:** 3 good
**Presentation:** 3 good
**Contribution:** 2 fair

**Summary:**

The authors propose a framework for convergence analysis of unrolled differentiation of quadratic objectives. To that end, they use a connection between optimization methods and residual polynomials which allows one to establish convergence bounds by evaluating appropriate polynomials on the spectrum of the hessian. From their theoretical results it can be seen that the convergence behaviour of the Jacobian of the argmin and the optimization objective behave differently during unrolling. A larger step size that yields fastest convergence of the optimization objective induces in turn a longer burn-in phase of the Jacobian sequence. Smaller step sizes can mitigate this burn-in phase while inducing slower asymptotic convergence. This tradeoff is called the curse of unrolling by the authors. Finally the authors propose an accelerated unrolling procedure for quadratic objectives that is motivated by the theory of Sobolev Polynomials and is shown to outperform reference methods (GD and Chebyshew) on different datasets.

**Questions:**

Some typos I found:

17: function(s)

27: differentiation(m)

30: of (an) iterative algorithm

32: What does "mode differentiation" stand for here?

49: algorithm(s)

189: iteration(s)

194: (that) that

217: (we) don't have

Top of page 8: Figure ??

260: there is no Figure 5

280: should it be "formalism does (not) apply"?

One thing I was wondering about is whether the analysis still works when $n > 1$, i.e., when $\theta$ is not a scalar but a vector. How do I need to read for example (2) in that case? What is the dimension of $\partial H(\theta)$ then?

**Limitations:**

Yes.

**Strengths And Weaknesses:**

From my perspective, the presented results are novel and technically correct. Apart from a few typos and missing references that can likely be fixed, I also find that the paper is well-structured, clearly written, and has some comprehensive figures. I personally think that the theoretical underpinning of distinct convergence phases during unrolling is a very interesting result. Also I would say that the methodology and the derived accelerated method are strengths of the paper. Clear limitations that the authors also mention in their discussion are that the results only apply to first-order methods and quadratic objectives. With that said, the presented results are in my opinion still interesting, although maybe less relevant to a broader audience.

---

> ### Author Response · Authors · 2022-08-02
> **Response to reviewer**
>
>
> We thank the reviewer for the time spent on the paper and their insightful questions.
>
> ---
>
> ***Q:** One thing I was wondering about is whether the analysis still works when $n>1$, i.e., when $\theta$ is not a scalar but a vector. How do I need to read for example (2) in that case? What is the dimension of $\partial H(\theta)$ then?*
>
> **A:** We do not need to assume that $n=1$. In the case $n>1$, we have that $\partial H(\theta)$ is a tensor of size $n \times d \times d$. The multiplication between $\partial H(\theta)$ and $H(\theta)$ corresponds to $n$ matrix multiplication along each of the $n$ first coordinates of $\partial H(\theta)$, i.e., if we call $\partial H(\theta) H=T$, we have $ T_{ijk}=[\partial_{\theta_i} H(\theta) H]_{jk}$
>
>
> An example of such a quadratic function $f(x,\theta)$ with $d=n>1$ would be
> $f(x,\theta) := \frac12 (\|Ax-b\|^2 + \langle\theta,x\rangle^2).$
> In that case we have $H(\theta) = A^\top A + \theta \theta^\top$ and thus $\partial H(\theta) = (\bar\theta_1 \theta^\top + \theta \bar\theta_1^\top, \ldots, \bar\theta_n \theta^\top + \theta \bar\theta_n^\top)$ where $\bar\theta_i \in \mathbb{R}^n$ denotes the vector with null coordinates expect of the index $i$ for which the coordinate is the $i^{th}$ coordinate of $\theta$.
>
> In that case, we have for any matrices $A, B \in \mathbb{R}^{d \times d}$,
> $A\partial H(\theta)B = (A(\bar\theta_1 \theta^\top + \theta \bar\theta_1^\top)B, \ldots, A(\bar\theta_n \theta^\top + \theta \bar\theta_n^\top)B) \in \mathbb{R}^{n \times d \times d} $
>
> We've updated Assumption 2 in the manuscript with the more precise definition above.
>
> ---
>
> **Remark:** *Apart from a few typos and missing references that can likely be fixed*
>
> **A:** We have also corrected the typos mentioned. Thanks!
>
> ---
>
> **Limitations: the results only apply to quadratic objectives**.
>
> **A**: We have now added experiments on a logistic regression objective to Appendix B. The figure shows the same pattern as Figure 2, showing that the two-phase dynamics extends to other objectives beyond quadratics.

---

### Official Review · Reviewer_9FSi · 2022-07-12

**Rating:** 8
**Confidence:** 3
**Soundness:** 3 good
**Presentation:** 3 good
**Contribution:** 3 good

**Summary:**

This paper pointed out an interesting phenomenon in bi-level optimization: the tradeoff between the convergence rate and the length of "burn-in" phase. Mathematical descriptions and analysis are provided.

----
The authors addressed my concerns and I updated my score.

**Questions:**

1. At the beginning of this paper, Fig. 1 shows the difference between the function suboptimality and Jacobian suboptimality. This phenomenon is really interesting. However, in the main text, there is only the upper bound for Jacobian suboptimality. Why is there no bounds for the function gap? I believe that this paper will be better if the function gap bound can be provided and it matches Fig. 1.
2. Through this paper, only quadratic functions are considered both in the theoretical and experimental parts. Is it possible to provide some experiments to show that the phenomenon "curse of unrolling" exists not only with quadratic functions but also with more general functions? At least some references on this phenomenon should be provided. If such a phenomenon only happens on quadratic functions, the significance of this paper will be weakened.
3. Line 182: typo: "Figure ??" Such typo also occurs in the caption of Figure 3.
4. Figure 4: Why does gradient descent on "synthetic dataset" diverge?

**Limitations:**

My main concerns are Questions 1 & 2 in the previous section, some potential suggestions are provided following the questions.
This paper is purely theoretical and I think it has no potential negative societal impact.

**Strengths And Weaknesses:**

The "curse of unrolling" and its mathematical descriptions are really interesting. Thus, I think this paper is very insightful. But the completeness and presentation have room to improve. I listed some questions below and I'm open to updating the grade if the authors are able to fix my concerns.

---

> ### Author Response · Authors · 2022-08-02
> **Response to reviewer**
>
>
> We want to thank the Reviewer for the time spent on the paper and for their insightful questions. Below is our answer to the Reviewer's comment, and we hope we have addressed and resolved all of their concerns. The Reviewer can also find the requested experiments in the appendices of the article.
>
> ---
> ***Q1:** Fig. 1 shows the difference between the function suboptimality and Jacobian suboptimality. In the main text, there is only the upper bound for Jacobian suboptimality. Why is there no bounds for the function gap?*
>
> **A:** The bounds for the function gap for the methods considered in Figure 1 are already known as they are the ones of standard first-order methods. It is known that the convergence rate of these methods is linear depending on the condition number. For example, the bound on the distance to the solution of gradient descent reads
>
> $\|x_t-x_\star\|^2 \leq \|x_0-x_\star\|^2 \left(1-h\frac{2\mu L}{\mu+L}\right)^t,\quad h<2/L,$
>
> Where $h$ is the step size of gradient descent, and $\mu, L$ are respectively the strong convexity and smoothness constant. A similar bound can be derived for the function values (e.g., Theorem 2.1.15 in Nesterov Y., *Introductory lectures on convex optimization: A basic course*. Springer Science & Business Media, 2003).
>
> ---
>
>
> ***Q2:** Is it possible to provide some experiments to show that the phenomenon "curse of unrolling" exists not only with quadratic functions but also with more general functions?*
>
> **A:** We have added an experiment illustrating the curse of unrolling on a problem with a logistic regression loss in the Appendices (Experiments on Logistic regression). The figure shows the same pattern as Figure 2 for logistic regression.
>
> The only significant difference with the least squares loss is the range of step-size values that exhibit the initial burn-in phase. While for the quadratic loss, these are step-sizes close to $h=\frac{2}{L + \mu}$, in the case of logistic regression, $L$ is a crude global upper bound, and so this step-size is not necessarily the one that achieves the fastest convergence rate. The featured two-phase curve was computed using the step size with the fastest asymptotic rate, calculated through a grid search on the step-size values.
>
> ---
>
> ***Q3:** Line 182: typo: "Figure ??" Such typo also occurs in the caption of Figure 3.*
>
> **A:** We've corrected that now, thanks.
>
> ---
>
> ***Q4:** Figure 4: Why does gradient descent on "synthetic dataset" diverge?*
>
> **A:** Gradient descent diverges in this figure due to its long burn-in period. Note that in Figure 2, we have used a different $x$ scale to be able to show both phases. We've observed that in these datasets, the burn-in phase of gradient descent is *ten times longer* than that of Chebyshev. We can only see the early divergent phase of gradient descent when both are overlaid.

---

### Author Response · Authors · 2022-08-02
**Revised manuscript**

We would like to thank the anonymous reviewers for the time spent on the paper and their insightful comments. We have uploaded a revised manuscript. The main difference with respect to the original manuscript is the addition of experiments on logistic regression (Appendix B),  and the corrections and clarifications suggested by the reviewers. The supplementary material contains a revision_diff.pdf document showing the differences with respect to the original submission.

---

### Meta-Review · Area_Chair_JT5a · 2022-08-26

**Recommendation:** Accept
**Confidence:** Certain

**Metareview:**

All the reviewers judged the paper to be novel and interesting and voted to accept it. There were some concerns about the size of the experiments that were partially solved during the discussion phase. Hence, I encourage the authors to add even more experiments in the camera ready to increase the impact of the paper.

**Award:**

No

---

### Decision · Program_Chairs · 2022-09-14

Accept